# Sequential Reptile: Inter-Task Gradient Alignment for Multilingual Learning

**Seanie Lee**[1*], **Hae Beom Lee**[1*], **Juho Lee**[1,2], **Sung Ju Hwang**[1,2]
KAIST[1], AITRICS[2], South Korea
{lsnfamily02, haebeom.lee , juholee, sjhwang82}@kaist.ac.kr

## Abstract

Multilingual models jointly pretrained on multiple languages have achieved remarkable performance on various multilingual downstream tasks. Moreover, models finetuned on a single monolingual downstream task have shown to generalize to unseen languages. In this paper, we first show that it is crucial for those tasks to align gradients between them in order to maximize knowledge transfer while minimizing negative transfer. Despite its importance, the existing methods for gradient alignment either have a completely different purpose, ignore inter-task alignment, or aim to solve continual learning problems in rather inefficient ways. As a result of the misaligned gradients between tasks, the model suffers from severe negative transfer in the form of catastrophic forgetting of the knowledge acquired from the pretraining. To overcome the limitations, we propose a simple yet effective method that can efficiently align gradients between tasks. Specifically, we perform each inner-optimization by sequentially sampling batches from all the tasks, followed by a Reptile outer update. Thanks to the gradients aligned between tasks by our method, the model becomes less vulnerable to negative transfer and catastrophic forgetting. We extensively validate our method on various multi-task learning and zero-shot cross-lingual transfer tasks, where our method largely outperforms all the relevant baselines we consider.

## 1 Introduction

Multilingual language models (Devlin et al., 2019; Conneau & Lample, 2019; Conneau et al., 2020; Liu et al., 2020; Lewis et al., 2020a; Xue et al., 2021) have achieved impressive performance on a variety of multilingual natural language processing (NLP) tasks. Training a model with multiple languages jointly can be understood as a multi-task learning (MTL) problem where each language serves as a distinct task to be learned (Wang et al., 2021). The goal of MTL is to make use of relatedness between tasks to improve generalization without negative transfer (Kang et al., 2011; Kumar & Daumé III, 2012; Lee et al., 2016; Wang et al., 2019b; 2020b). Likewise, when we train with a downstream multilingual MTL objective, we need to maximize knowledge transfer between the languages while minimizing negative transfer between them. This is achieved by developing an effective MTL strategy that can prevent the model from memorizing task-specific knowledge not easily transferable across the languages.

Such MTL problem is highly related to the gradient alignment between the tasks, especially when we finetune a well-pretrained model like multilingual BERT (Devlin et al., 2019). We see from the bottom path of Fig. 1 that the cosine similarity between the task gradients (gradients of MTL losses individually computed for each task) tend to gradually decrease as we finetune the model with the MTL objective. It means that the model gradually starts memorizing task-specific (or language-specific) knowledge not compatible across the languages, which can cause a negative transfer from one language to another. In case of finetuning the well-pretrained model, we find that it causes catastrophic forgetting of the pretrained knowledge. Since the pretrained model is the fundamental knowledge shared across all NLP tasks, such catastrophic forgetting can severely degrade the performance of all tasks. Therefore, we want our model to maximally retain the knowledge of the pretrained model by finding a good trade-off between minimizing the downstream MTL loss

---

*Equal contribution

Figure 1: **Concepts.** Black arrows denote finetuning processes. The darker the part of the arrows, the lower the MTL loss. Upper and bottom path shows better and worse trade-off, respectively. Colored arrows denote task gradients. Blue and red color shows high and low cosine similarity, respectively. We demonstrate this concept with the actual experimental results in Fig. 7a.

and maximizing the cosine similarity between the task gradients, as illustrated in the upper path of Fig. 1. In this paper, we aim to solve this problem by developing an MTL method that can efficiently align gradients across the tasks while finetuning the pretrained model.

There has been a seemingly related observation by Yu et al. (2020) that the conflict (negative cosine similarity) between task gradients makes it hard to optimize the MTL objective. They propose to manually alter the direction of task gradients whenever the task gradients conflict to each other. However, their intuition is completely different from ours. They *manually* modify the task gradients whenever the gradient conflicts happen, which leads to more aggressive optimization of MTL objective. In case of finetuning a well-pretrained model, we find that it simply leads to catastrophic forgetting. Instead, we aim to make the model converge to a point where task gradients are naturally aligned, leading to less aggressive optimization of the MTL objective (See the upper path of Fig. 1).

Then a natural question is if we can alleviate the catastrophic forgetting with early stopping. Our observation is that whereas early stopping can slightly increase cosine similarity to some extent, it is not sufficient to find a good trade-off between minimizing MTL objective and maximizing cosine similarity to improve generalization (See Fig. 1). It means that we may need either an implicit or explicit objective for gradient alignment between tasks. Also, Chen et al. (2020) recently argue that we can mitigate catastrophic forgetting by adding $\ell_2$ regularization to AdamW optimizer (Loshchilov & Hutter, 2019). They argue that the resultant optimizer penalizes $\ell_2$ distance from the pretrained model during the finetuning stage. However, unfortunately we find that their method is not much effective in preventing catastrophic forgetting in the experimental setups we consider.

On the other hand, Reptile (Nichol et al., 2018) implicitly promotes gradient alignment between mini-batches *within a task*. Reptile updates a shared initial parameter individually for each task, such that the task gradients are not necessarily aligned across the tasks. In continual learning area, MER (Riemer et al., 2019) and La-MAML (Gupta et al., 2020) propose to align the gradients between sequentially incoming tasks in order to maximally share the progress on their objectives. However, as they focus on continual learning problems, they require explicit memory buffers to store previous task examples and align gradients with them, which is complicated and costly. Further, their methods are rather inefficient in that the inner-optimization is done with batch size set to 1, which takes significantly more time than usual batch-wise training. Therefore, their methods are not straightforwardly applicable to multilingual MTL problems we aim to solve in this paper.

In this paper, we show that when we finetune a well-pretrained model, it is sufficient to align gradients between the currently given downstream tasks in order to retain the pretrained knowledge, without accessing the data used for pretraining or memory buffers. Specifically, during the finetuning stage, we sequentially sample mini-batches *from all the downstream tasks* at hand to perform a single inner-optimization, followed by a Reptile outer update. Then, we can efficiently align the gradients between tasks based on the implicit dependencies between the inner-update steps. This procedure, which we call *Sequential Reptile*, is a simple yet effective method that can largely improve the performance of various downstream multilingual tasks by preventing negative transfer and catastrophic forgetting in an efficient manner. We summarize our contributions as follows.

- We show that when finetuning a well-pretrained model, gradients not aligned between tasks can cause negative transfer and catastrophic forgetting of the knowledge acquired from the pretraining.

- To solve the problem, we propose *Sequential Reptile*, a simple yet effective MTL method that efficiently aligns gradients between tasks, thus prevents negative transfer and catastrophic forgetting.
- We extensively validate our method on various MTL and zero-shot cross-lingual transfer tasks, including question answering, named entity recognition and natural language inference tasks, in which our method largely outperforms all the baselines.

## 2    RELATED WORKS

**Multi-task Learning**    The goal of MTL is to leverage relatedness between tasks for effective knowledge transfer while preventing negative interference between them (Zhang & Yeung, 2010; Kang et al., 2011; Lee et al., 2016). GradNorm (Chen et al., 2018) tackles task imbalance problem by adaptively weighting each task loss. Another line of literature propose to search Pareto optimal solutions which represent trade-offs between the tasks (Sener & Koltun, 2018; Lin et al., 2019). Recently, Yu et al. (2020) and Wang et al. (2021) propose to manually resolve the conflict between task gradients to more aggressively optimize the given MTL objective. However, their goal is completely different from ours because here we focus on preventing negative transfer by finding a model that can naturally align the task gradients without such manual modifications.

**Multilingual Language Model**    Training a multilingual language model is a typical example of multi-task learning. Most of the previous works focus on jointly pretraining a model with hundreds of languages to transfer common knowledge between the languages (Devlin et al., 2019; Conneau & Lample, 2019; Conneau et al., 2020; Liu et al., 2020; Lewis et al., 2020a; Xue et al., 2021). Some literature show the limitation of jointly training the model with multilingual corpora (Arivazhagan et al., 2019; Wang et al., 2020b). Several follow-up works propose to tackle the various accompanying problems such as post-hoc alignment (Wang et al., 2019c; Cao et al., 2019), data balancing (Wang et al., 2020a) and loss curvature-aware optimization to improve the performance of low resource languages (Li & Gong, 2021). In this paper, we focus on how to finetune a well pretrained multilingual language model by preventing catastrophic forgetting of the pretrained knowledge.

**Zero-shot Cross Lingual Transfer**    Zero-shot cross-lingual transfer is to train a model with monolingual labeled data and evaluate it on some unseen target languages without further finetuning the model on the target languages. Nooralahzadeh et al. (2020) utilize meta-learning to learn how to transfer knowledge from high resource languages to low resource ones. Hu et al. (2021) and Pan et al. (2021) leverage a set of paired sentences from different languages to train the model and minimize the distance between the representation of the paired sentences. Instead, we partition the monolingual labeled data into groups and consider them as a set of tasks for multi-task learning.

## 3    APPROACH

The goal of multi-task learning (MTL) is to estimate a model parameter $\phi$ that can achieve good performance across all the given $T$ tasks, where each task $t = 1, \dots, T$ has task-specific data $\mathcal{D}_t$. We learn $\phi$ by minimizing the sum of task losses.

$$\min_{\phi} \sum_{t=1}^{T} \mathcal{L}(\phi; \mathcal{D}_t) + \lambda \Omega(\phi) \tag{1}$$

where $\Omega(\phi)$ is a regularization term and $\lambda \geq 0$ is an associated coefficient.

**Reptile**    We briefly review Reptile (Nichol et al., 2018), an efficient first-order meta-learning method suitable for large-scale learning scenario. We show that Reptile has an approximate learning objective of the form in Eq. 1. Although Reptile is originally designed for learning a shared initialization, we can use the initialization $\phi$ for actual predictions without any adaptation (Riemer et al., 2019). Firstly, given a set of $T$ tasks, we *individually* perform the task-specific optimization from $\phi$. Specifically, for each task $t = 1, \dots, T$, we perform the optimization by sampling mini-batches $\mathcal{B}_t^{(1)}, \dots, \mathcal{B}_t^{(K)}$ from task data $\mathcal{D}_t$ and taking gradient steps with them.

$$\theta_t^{(0)} = \phi, \qquad \theta_t^{(k)} = \theta_t^{(k-1)} - \alpha \frac{\partial \mathcal{L}(\theta_t^{(k-1)}; \mathcal{B}_t^{(k)})}{\partial \theta_t^{(k-1)}} \tag{2}$$

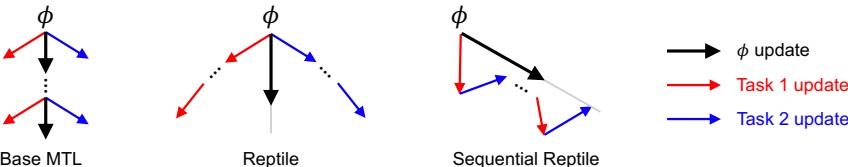

Figure 2: Comparison between the methods.

for $k = 1, \ldots, K$, where $\alpha$ is an inner-learning rate and $\theta_t^{(k)}$ denotes the task-specific parameter of task $t$ evolved from $\phi$ by taking $k$ gradient steps. After performing $K$ gradient steps for all $T$ tasks, we meta-update $\phi$ as follows:

$$\phi \leftarrow \phi - \eta \cdot \frac{1}{T} \sum_{t=1}^{T} \text{MG}_t(\phi), \quad \text{where} \quad \text{MG}_t(\phi) = \phi - \theta_t^{(K)} \tag{3}$$

where $\eta$ denotes an outer-learning rate. Nichol et al. (2018) show that expectation of $\text{MG}_t(\phi)$ over the random sampling of batches, which is the meta-gradient of task $t$ evaluated at $\phi$, can be approximated as follows based on Taylor expansion:

$$\mathbb{E}\left[\text{MG}_t(\phi)\right] \approx \frac{\partial}{\partial \phi} \mathbb{E}\left[\sum_{k=1}^{K} \mathcal{L}(\phi; \mathcal{B}_t^{(k)}) - \frac{\alpha}{2} \sum_{k=1}^{K} \sum_{j=1}^{k-1} \left\langle \frac{\partial \mathcal{L}(\phi; \mathcal{B}_t^{(k)})}{\partial \phi}, \frac{\partial \mathcal{L}(\phi; \mathcal{B}_t^{(j)})}{\partial \phi} \right\rangle \right] \tag{4}$$

where $\langle \cdot, \cdot \rangle$ denotes a dot product. We can see that $\mathbb{E}[\text{MG}_t(\phi)]$ approximately minimizes the task-specific loss (first term) and maximizes the inner-products between gradients computed with different batches (second term). The inner-learning rate $\alpha$ controls the trade-off between them. However, the critical limitation is that the dot product does not consider aligning gradients computed from different tasks. This is because each inner-learning trajectory consists of the batches $\mathcal{B}_t^{(1)}, \ldots, \mathcal{B}_t^{(K)}$ sampled from the same task data $\mathcal{D}_t$.

## 3.1 SEQUENTIAL REPTILE

In order to consider gradient alignment *across tasks* as well, we propose to let the inner-learning trajectory consist of mini-batches randomly sampled *from all tasks*, which we call *Sequential Reptile*.

Unlike Reptile where we run $T$ task-specific inner-learning trajectory (in parallel), now we have a single learning trajectory responsible for all $T$ tasks (See Figure 2). Specifically, for each inner-step $k$, we randomly sample a task index $t_k \in \{1, \ldots, T\}$ and a corresponding mini-batch $\mathcal{B}_{t_k}^{(k)}$, and then sequentially update $\theta^{(k)}$ as follows.

$$\theta^{(0)} = \phi, \qquad \theta^{(k)} = \theta^{(k-1)} - \alpha \frac{\partial \mathcal{L}(\theta^{(k-1)}; \mathcal{B}_{t_k}^{(k)})}{\partial \theta^{(k-1)}}, \quad \text{where} \quad t_k \sim \text{Cat}(p_1, \ldots, p_T) \tag{5}$$

$\text{Cat}(p_1, \ldots, p_T)$ is a categorical distribution parameterized by $p_1, \ldots, p_T$, the probability of each task to be selected. For example, we can let $p_t \propto (N_t)^q$ where $N_t$ is the number of training instances for task $t$ and $q$ is some constant. After $K$ gradient steps with Eq. 5, we update $\phi$ as follows

$$\phi \leftarrow \phi - \eta \cdot \text{MG}(\phi), \quad \text{where} \quad \text{MG}(\phi) = \phi - \theta^{(K)} \tag{6}$$

Again, based on Taylor expansion, we have the following approximate form for expectation of the meta-gradient $\text{MG}(\phi)$ over the random sampling of tasks (See derivation in Appendix C).

$$\mathbb{E}\left[\text{MG}(\phi)\right] \approx \frac{\partial}{\partial \phi} \mathbb{E}\left[\sum_{k=1}^{K} \mathcal{L}(\phi; \mathcal{B}_{t_k}^{(k)}) - \frac{\alpha}{2} \sum_{k=1}^{K} \sum_{j=1}^{k-1} \left\langle \frac{\partial \mathcal{L}(\phi; \mathcal{B}_{t_k}^{(k)})}{\partial \phi}, \frac{\partial \mathcal{L}(\phi; \mathcal{B}_{t_j}^{(j)})}{\partial \phi} \right\rangle \right] \tag{7}$$

Note that the critical difference of Eq. 7 from Eq. 4 is that the dot product between the two gradients is computed from the different tasks, $t_k$ and $t_j$. Such inter-task dependency appears as we randomly sample batches from all the tasks sequentially and compose a single learning trajectory with them. As a result, Eq. 7, or Sequential Reptile promotes the gradient alignments between the tasks, preventing the model from memorizing language specific knowledge. It also means that the model can find a good trade-off between minimizing the MTL objective and inter-task gradient alignment, thereby effectively preventing catastrophic forgetting of the knowledge acquired from pretraining.

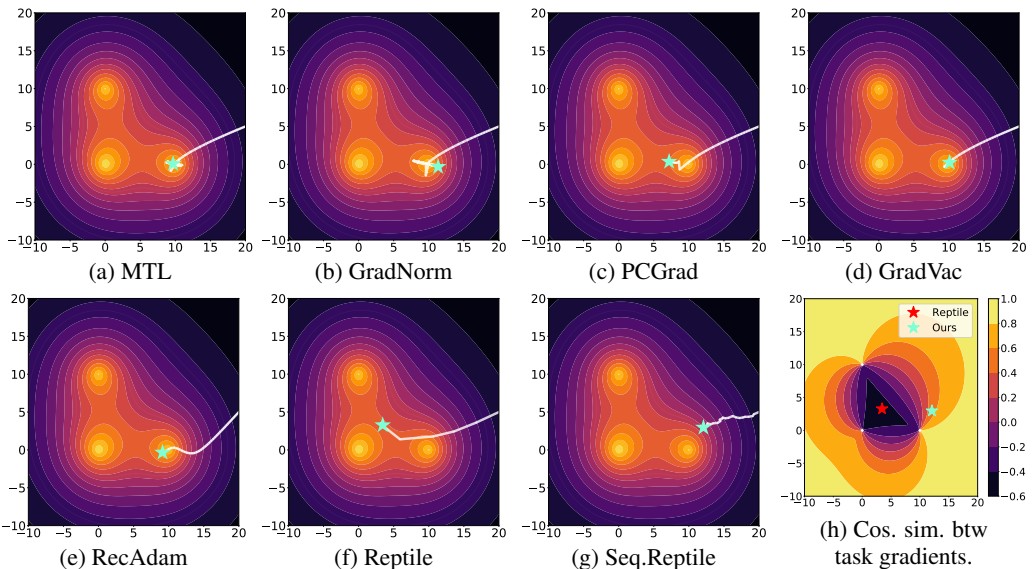

Figure 3: **(a)~(g)** Loss surface and learning trajectory of each method. **(h)** Heatmap shows average pair-wise cosine similarity between the task gradients.

## 4 EXPERIMENTS

We first verify our hypothesis with synthetic experiments. We then validate our method by solving multi-task learning (MTL) and zero-shot cross-lingual transfer tasks with large-scale real datasets.

**Baselines**    We compare our method against the following relevant baselines.

1. **STL:** Single Task Learning (STL) model trained on each single language.
2. **MTL:** Base MTL model of which objective is Eq. 1 without the regularizer $\Omega(\phi)$.
3. **GradNorm (Chen et al., 2018):** This model tackles task imbalance problem in MTL. It prevents the training from being dominated by a single task by adaptively weighting each task gradient.
4. **PCGrad (Yu et al., 2020):** This model aims to optimize MTL objective more aggressively by resolving gradient conflicts. Specifically, it projects a task gradient onto the other task gradients if the inner product between them is negative.
5. **GradVac (Wang et al., 2021):** Similarly to PCGrad, this model alters the task gradients to match the empirical moving average of cosine similarity between the task gradients.
6. **RecAdam (Chen et al., 2020):** A model trained with RecAdam optimizer to prevent catastrophic forgetting by penalizing $\ell_2$ distance from the the pretrained model.
7. **Reptile (Nichol et al., 2018):** A first-order meta-learning method suitable for large-scale learning scenario. Unlike our method, Reptile performs inner-optimization individually for each task.
8. **Sequential Reptile:** Our method that can align gradients across the tasks by composing the inner-learning trajectory with all the tasks.

### 4.1 SYNTHETIC EXPERIMENTS

We first verify our hypothesis with the following synthetic experiments. We define three local optima $x_1 = (0, 10), x_2 = (0, 0)$, and $x_3 = (10, 0)$ in a 2-dimensional space. Then, we define three tasks as minimizing each of the following loss functions w.r.t $\phi \in \mathbb{R}^2$.

$$\mathcal{L}_i(\phi) = -200 \cdot \exp\left(0.2 \cdot \|\phi - x_i\|_2\right), \quad \text{for} \quad i = 1, 2, 3.$$

MTL objective is defined as $\sum_{i=1}^{3} \mathcal{L}_i(\phi)$, which we optimize from the initialization $(20, 5)$.

**Results and analysis**    Fig. 3 shows the MTL loss surface and the learning trajectory of each method. We observe that except for Reptile and Sequential Reptile, all the other baselines converge to one of the MTL local minima, failing to find a reasonable solution that may generalize better across the tasks. While Reptile can avoid such a minimum, the resultant solution has very low cosine similarity (See Fig. 3h) because it does not enforce gradient alignments between tasks. On the other hand, Figure 3h shows that our Sequential Reptile tends to find a reasonable trade-off

between minimizing MTL loss and maximizing cosine similarity, thanks to the implicit enforcement of inter-task gradient alignment in Eq. 7.

Table 1: F1 and EM score on the TYDI-QA dataset for QA. The best result for multilingual models is marked with bold while the underline denotes the best result among all the models including the monolingual model.

| | Question Answering (F1/EM) | | | | | | | | | |
|---|---|---|---|---|---|---|---|---|---|---|
| Method | ar | bn | en | fi | id | ko | ru | sw | te | Avg. |
| STL | 80.5 / 65.9 | 70.9 / 58.4 | 72.0 / 59.2 | 76.2 / 63.7 | 82.7 / 70.6 | 61.0 / 50.8 | 73.4 / 56.5 | 78.4 / 70.1 | 81.1 / 66.4 | 75.1 / 62.4 |
| MTL | 79.7 / 64.7 | 74.7 / 64.0 | 72.8 / 61.1 | 77.8 / 64.7 | 82.9 / 71.3 | 64.0 / 53.4 | 73.9 / 57.1 | 80.5 / 72.5 | 82.5 / 67.9 | 76.5 / 64.1 |
| RecAdam | 79.0 / 63.7 | 72.5 / 62.4 | 73.5 / 62.8 | 76.9 / 64.9 | 82.1 / 72.2 | 64.6 / 54.5 | 73.9 / 57.6 | 80.4 / 72.9 | 82.8 / 68.4 | 76.2 / 64.4 |
| GradNorm | 78.8 / 62.9 | 72.5 / 61.9 | 73.4 / 60.6 | 78.5 / 65.8 | 83.7 / 74.3 | 66.2 / 53.9 | 74.5 / 57.7 | 80.7 / 72.7 | **83.3** / **68.9** | 76.8 / 64.3 |
| PCGrad | 79.9 / 65.0 | 72.6 / 61.5 | 74.6 / 62.3 | 78.3 / 65.7 | 82.7 / 73.0 | 65.9 / 56.1 | 74.2 / 57.3 | 80.6 / 73.0 | 82.5 / 68.1 | 76.8 / 64.7 |
| GradVac | 80.1 / 64.7 | 71.5 / 59.2 | 73.0 / 61.2 | 78.6 / 65.5 | 83.1 / 72.4 | 63.8 / 53.4 | 74.1 / 57.6 | 80.6 / 72.6 | 82.2 / 67.5 | 76.3 / 63.8 |
| Reptile | 79.8 / 65.2 | __75.1__ / __64.1__ | 74.0 / 62.8 | 78.8 / 65.3 | 83.8 / 73.6 | 65.7 / 55.9 | 75.5 / 58.7 | 81.4 / 72.9 | 83.1 / 68.5 | 77.5 / 65.2 |
| Seq.Reptile | __81.2__ / __66.7__ | 73.9 / 62.6 | __76.7__ / __65.2__ | __79.4__ / __66.3__ | __84.9__ / __74.7__ | __68.0__ / __58.2__ | __76.8__ / __59.2__ | __82.9__ / __74.8__ | 83.0 / 68.6 | __78.5__ / __66.3__ |

Table 2: F1 score on WikiAnn dataset for NER. The best result for multilingual models is marked with bold while the underline denotes the best result among all the models including the monolingual model.

| | Named Entity Recognition (F1) | | | | | | | | | | | |
|---|---|---|---|---|---|---|---|---|---|---|---|---|
| Method | de | en | es | hi | jv | kk | mr | my | sw | te | tl | yo | Avg. |
| STL | __90.3__ | __85.0__ | __92.0__ | __89.7__ | 59.1 | 88.5 | __89.4__ | 61.7 | 90.7 | 80.1 | __96.3__ | 77.7 | 83.3 |
| MTL | 83.4 | 77.8 | 87.6 | 82.3 | 77.7 | 87.5 | 82.2 | 75.7 | 87.5 | 78.8 | 83.5 | 90.8 | 82.9 |
| RecAdam | 84.5 | 80.0 | 88.5 | 82.7 | **85.3** | 88.5 | 84.4 | 70.3 | 89.0 | 81.6 | 87.7 | 91.6 | 84.5 |
| GradNorm | 83.6 | 77.5 | 87.3 | 82.8 | 78.3 | 87.8 | 81.3 | 73.5 | 85.4 | 78.9 | 83.6 | 91.4 | 82.6 |
| PCGrad | 83.8 | 78.5 | 88.1 | 81.7 | 79.7 | 87.8 | 81.7 | 74.4 | 85.9 | 78.4 | 85.7 | 92.3 | 83.1 |
| GradVac | 83.9 | 79.5 | 88.3 | 81.8 | 80.6 | 87.5 | 82.2 | 73.9 | 87.9 | 79.4 | 87.9 | __93.0__ | 83.8 |
| Reptile | 85.9 | 82.4 | 90.0 | 86.3 | 81.3 | 81.4 | 86.8 | 61.8 | 90.6 | 72.7 | 92.8 | __93.0__ | 83.7 |
| Seq.Reptile | **87.4** | **83.9** | **90.8** | **88.1** | 85.2 | __89.4__ | **88.9** | __76.0__ | __91.5__ | __82.5__ | **94.7** | 92.5 | __87.5__ |

## 4.2 MULTI-TASK LEARNING

We next verify our method with large-scale real datasets. We consider multi-task learning tasks such as multilingual Question Answering (QA) and Named Entity Recognition (NER). For QA, we use "Gold passage" of TYDI-QA (Clark et al., 2020) dataset where a QA model predicts a start and end position of answer from a paragraph for a given question. For NER, we use WikiAnn dataset (Pan et al., 2017) where a model classifies each token of a sentence into three classes. We consider each language as a distinct task and train MTL models.

**Implementation Details** For all the experiments, we use multilingual BERT (Devlin et al., 2019) base model as a backbone network. We fintune it with AdamW (Loshchilov & Hutter, 2019) optimizer, setting the inner-learning rate $\alpha$ to $3 \cdot 10^{-5}$. We use batch size 12 for QA and 16 for NER, respectively. For our method, we set the outer learning rate $\eta$ to 0.1 and the number inner-steps $K$ to 1000. Following Wang et al. (2021), for all the baselines, we sample eight tasks proportional to $p_t \propto (N_t)^{1/5}$ where $N_t$ is the number of training instances for task $t$. For our Sequential Reptile, we set the parameter of the categorical distribution in Eq. 5 to the same $p_t$.

**Results** We compare our method, Sequential Reptile against the baselines on QA and NER tasks. Table 1 shows the results of QA task. We see that our method outperforms all the baselines including STL on most of the languages. Interestingly, all the other baselines but ours underperform STL on Arabic language which contains the largest number of training instances ($2 \sim 3$ times larger than the other languages. See Appendix B for more information about data statistics). It implies that the baselines suffer from negative transfer while ours is relatively less vulnerable. We can observe essentially the same tendency for NER task, which is a highly imbalanced such that the low-resource languages can have around 100 training instance, while the high-resource languages can have around 5,000 to 20,000 examples. Table 2 shows the results of NER task. We see that all the baselines but ours highly degrade the performance on high resource languages — de, en, es, hi, mr and tl, which means that they fail to address the imbalance problem properly and suffer from severe negative transfer. Even GradNorm is not effective in our experiments, which is developed to tackle the task imbalance problem by adaptively scaling task losses.

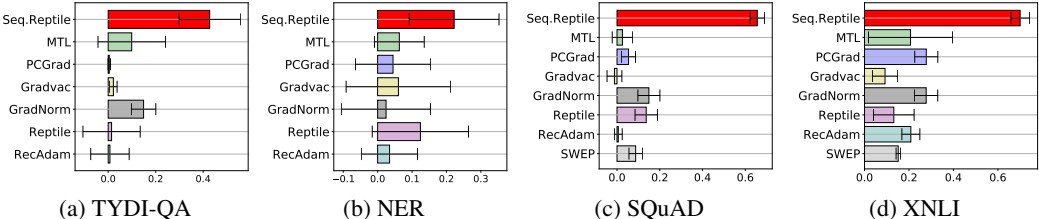

Figure 4: Average pair-wise cosine similarity between the gradients computed from different tasks.

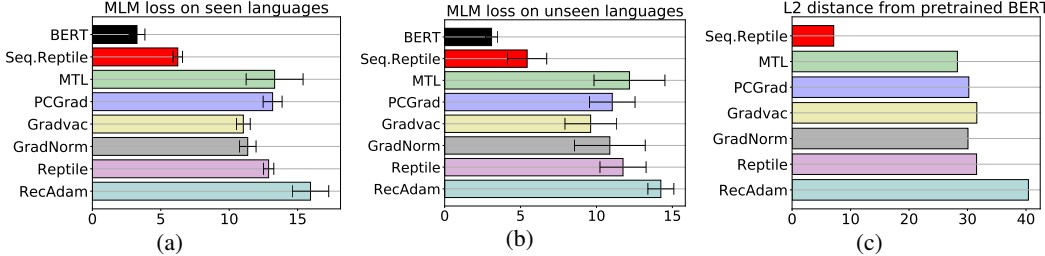

Figure 5: **(a, b)** Average MLM loss on **(a)** seen and **(b)** unseen languages from Common Crawl dataset. We mask 15% tokens of sentences from Common Crawl dataset, which is preprocessed and provided by Wenzek et al. (2020), and compute the masked language modeling loss (MLM), which is reconstruction loss of the masked sentences. **(c)** $\ell_2$ distance between the finetuned models and the initially pretrained BERT.

**Analysis** We next analyze the source of performance improvements. Our hypothesis is that our method can effectively filter out language specific knowledge when solving the downstream tasks, which prevents negative transfer and helps retain linguistic knowledge acquired from the pretraining.

Table 3: We evaluate the model, trained with TYDI-QA dataset, on five unseen languages from MLQA dataset.

| | | | TYDI-QA $\rightarrow$ MLQA (F1/EM) | | | |
|---|---|---|---|---|---|---|
| **Method** | **de** | **es** | **hi** | **vi** | **zh** | **Avg.** |
| **MTL** | 50.6 / 35.8 | 54.3 / 35.4 | 45.0 / 31.2 | 53.7 / 34.8 | 52.5 / 32.0 | 51.2 / 33.8 |
| **RecAdam** | 49.5 / 36.1 | 52.8 / 35.6 | 41.7 / 29.0 | 52.2 / 34.6 | 49.8 / 29.9 | 49.2 / 33.0 |
| **GradNorm** | 51.7 / 36.6 | 54.9 / 36.3 | 44.4 / 30.4 | 55.3 / 37.1 | 52.9 / 32.2 | 51.8 / 34.5 |
| **PCGrad** | 50.6 / 36.5 | 54.5 / 36.2 | 44.1 / 31.2 | 54.4 / 34.8 | 52.4 / 31.5 | 51.1 / 34.2 |
| **GradVac** | 50.0 / 35.4 | 53.0 / 35.2 | 41.6 / 28.6 | 52.9 / 34.9 | 51.3 / 31.2 | 49.8 / 33.1 |
| **Reptile** | 52.2 / 38.2 | 56.1 / 38.0 | 45.9 / 32.1 | 56.9 / 38.2 | 53.9 / 33.2 | 53.0 / 35.9 |
| **Seq.Reptile** | **53.7 / 38.5** | **57.6 / 38.9** | **47.7 / 33.7** | **58.1 / 39.2** | **55.1 / 34.7** | **54.4 / 37.0** |

Firstly, we quantitatively measure the cosine similarity between the gradients computed from different tasks in Fig. 4a (QA) and Fig. 4b (NER). We see from the figure that our Sequential Reptile shows the highest cosine similarity as expected, due to the approximate learning objective in Eq. 7. Such high cosine similarity between the task gradients implies that the model has captured the common knowledge well transferable across the languages. This is in contrast to the other baselines whose task gradients can have even negative cosine similarity with high probability. Such different gradient directions mean that the current model has memorized the knowledge not quite transferable across the languages, which can cause negative transfer from one language to others.

As a result of such negative transfer, we see from Fig. 5a and Fig. 5b that the baselines suffer from catastrophic forgetting. In those figures, high MLM losses on seen (Fig. 5a) and unseen languages (Fig. 5b) mean that the models have forgotten how to reconstruct the masked sentences, which is the original training objective of BERT. On the other hand, our method shows relatively lower MLM loss, demonstrating its effectiveness in preventing negative transfer and thereby alleviating the catastrophic forgetting. We further confirm this tendency in Fig. 5c by measuring the $\ell_2$ distance from the initially pretrained BERT model. We see that the distance is much shorter for our method than the baselines, which implies that ours can better retain the linguistic knowledge than the baselines.

We can actually test if the finetuned models have acquired knowledge transferable across the languages by evaluating on some unseen languages without further finetuning. In Table 3, we test the finetuned QA models on five unseen languages from MLQA dataset (Lewis et al., 2020b). Again,

Table 4: Zero-shot cross lingual transfer from English (SQuAD) to six unseen languages (MLQA).

| | \multicolumn{7}{c}{**SQuAD → MLQA** (F1/EM)} | | | | | | |
|---|---|---|---|---|---|---|---|
| **Method** | **ar** | **de** | **es** | **hi** | **vi** | **zh** | **Avg.** |
| **MTL** | 48.0 / 29.9 | 59.1 / 44.6 | 64.2 / 46.2 | 44.6 / 29.0 | 56.8 / 37.8 | 55.2 / 34.7 | 54.6 / 37.0 |
| **RecAdam** | 47.3 / 29.8 | 58.8 / 44.0 | 64.3 / 46.2 | 45.5 / 29.9 | 57.9 / 38.4 | 54.8 / 34.1 | 54.7 / 37.0 |
| **GradNorm** | 48.7 / 31.3 | 59.8 / 44.6 | 64.8 / 46.3 | 47.2 / 31.2 | 57.2 / 37.8 | 55.0 / 33.9 | 55.4 / 37.5 |
| **PCGrad** | 47.7 / 29.8 | 59.2 / 44.1 | 65.4 / 46.1 | 41.0 / 26.0 | 57.4 / 37.4 | 54.6 / 34.0 | 54.4 / 36.2 |
| **GradVac** | 45.3 / 28.3 | 58.2 / 43.4 | 63.9 / 45.9 | 42.4 / 27.9 | 56.3 / 37.1 | 53.0 / 32.9 | 53.1 / 35.9 |
| **SWEP** | 49.5 / 31.0 | 60.5 / **46.2** | 65.0 / 47.3 | 47.6 / 31.9 | 57.9 / 38.8 | 56.9 / 36.3 | 56.2 / 38.5 |
| **Reptile** | 46.8 / 29.9 | 58.7 / 44.4 | 65.1 / 47.5 | 41.5 / 27.9 | 56.1 / 37.7 | 53.9 / 33.9 | 53.6 / 36.8 |
| **Seq.Reptile** | **52.8 / 34.3** | **60.7** / 45.9 | **67.1 / 48.7** | **50.6 / 35.6** | **60.7 / 40.8** | 57.3 / 36.5 | **58.2 / 39.3** |

Table 5: Zero shot cross lingual transfer from English (MNLI) to fourteen unseen languages (XNLI).

| | \multicolumn{15}{c}{**MNLI → XNLI** (Accuracy)} | | | | | | | | | | | | | |
|---|---|---|---|---|---|---|---|---|---|---|---|---|---|---|
| **Method** | **ar** | **bg** | **de** | **el** | **es** | **fr** | **hi** | **ru** | **sw** | **th** | **tr** | **ur** | **vi** | **zh** | **Avg.** |
| **MTL** | 65.1 | 68.8 | 71.3 | 66.4 | 74.3 | 72.6 | 59.2 | 68.8 | 50.1 | 52.8 | 61.8 | 57.9 | 69.4 | 69.0 | 64.8 |
| **RecAdam** | 63.5 | 66.9 | 69.4 | 65.2 | 72.7 | 72.2 | 58.8 | 67.3 | 49.9 | 51.9 | 61.0 | 56.5 | 67.9 | 67.7 | 63.6 |
| **GradNorm** | 64.0 | 68.2 | 70.6 | 67.0 | 74.1 | 72.9 | 58.8 | 68.0 | 48.9 | 53.2 | 61.0 | 56.8 | 70.3 | 69.3 | 64.5 |
| **PCGrad** | 64.0 | 68.7 | 69.8 | 66.5 | 74.3 | 71.8 | 59.4 | 68.3 | 51.0 | 53.1 | 60.7 | 57.4 | 69.7 | 69.3 | 64.5 |
| **GradVac** | 62.3 | 67.8 | 69.2 | 65.9 | 72.6 | 72.2 | 59.6 | 67.1 | 51.1 | 52.9 | 61.7 | 56.4 | 68.8 | 68.0 | 63.9 |
| **SWEP** | 64.8 | **69.4** | 70.5 | 67.1 | 74.8 | 74.4 | 58.7 | **69.7** | 49.2 | 53.7 | 60.2 | 57.1 | 69.8 | 68.4 | 64.8 |
| **Reptile** | 63.3 | 66.6 | 69.4 | 64.9 | 72.6 | 71.3 | 58.3 | 66.9 | 46.4 | 47.5 | 58.6 | 55.9 | 68.6 | 66.9 | 62.6 |
| **Seq.Reptile** | **67.2** | 69.3 | **71.9** | **67.8** | **75.1** | 74.1 | **60.6** | 69.5 | **51.2** | **55.1** | **63.8** | **59.1** | 70.8 | **69.6** | **66.0** |

Sequential Reptile outperforms all the baselines. The results confirm that the gradient alignment between tasks is key for obtaining common knowledge transferable across the languages.

### 4.3 ZERO-SHOT CROSS LINGUAL TRANSFER

We next validate our method on zero-shot cross lingual transfer task, motivated from the previous results that our method can learn common knowledge well transferable across the languages. We train a model only with English annotated data and evaluate the model on target languages without further finetuning on the target datasets. To utilize the methods of MTL, we cluster the data into four groups with Gaussian mixture model. We focus on QA and natural language inference (NLI) task. For QA, we train the model on SQuAD (Rajpurkar et al., 2016) dataset and evaluate the model on six languages from MLQA dataset (Lewis et al., 2020b). For NLI, we use MNLI (Williams et al., 2018) dataset as a source training dataset and test the model on fourteen languages from XNLI (Conneau et al., 2018) as a target languages.

**Baselines** As well as the baselines from the previous experiments, we additionally include **SWEP** (Lee et al., 2021). It learns to perturb word embeddings and uses the perturbed input as extra training data, which is empirically shown to be robust against out-of distribution data.

**Implementation Detail** We finetune multilingual BERT-base model with AdamW. For QA, we use the same hyperparameter in multi-task learning experiment. For NLI, we use batch size 32 and choose the learning rate $3 \cdot 10^{-5}$ or $5 \cdot 10^{-5}$ with AdamW optimizer based on the performance on the validation set. For our model, we set the outer learning rate $\eta$ to 0.1 and the number inner steps $K$ to 1000. In order to construct multiple tasks from a single dataset, we cluster concatenation of questions and paragraphs from SQuAD or sentences from MNLI into four groups. Following Aharoni & Goldberg (2020), we encode the paragraphs or sentences into hidden representations with pretrained multilingual BERT. Then, we reduce the dimension of hidden representations with PCA and run Gaussian mixture model to partition them into disjoint four clusters. Since the number of training instances are almost evenly distributed for each task, we sample mini-batches from all the four tasks for all the baselines and set $p_t = 1/4$ for our model.

**Results and Analysis** The results of zero-shot cross lingual transfer show essentially the same tendency as those of multi-task learning in the previous subsection. Firstly, our model largely out-performs all the baselines for QA and NLI tasks as shown in Table 4 and 5. To see where the performance improvements come from, we compute the average pairwise cosine similarity between the gradients computed from different tasks in Fig. 4c and 4d. Again, Sequential Reptile shows much higher similarity than the baselines, which implies that our method can effectively filter out

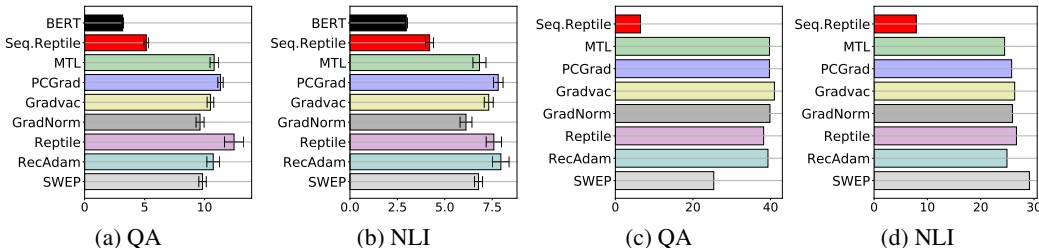

Figure 6: **(a,b)** Masked Language Modeling (MLM) loss. **(c,d)** $\ell_2$ distance from the pretrained BERT model.

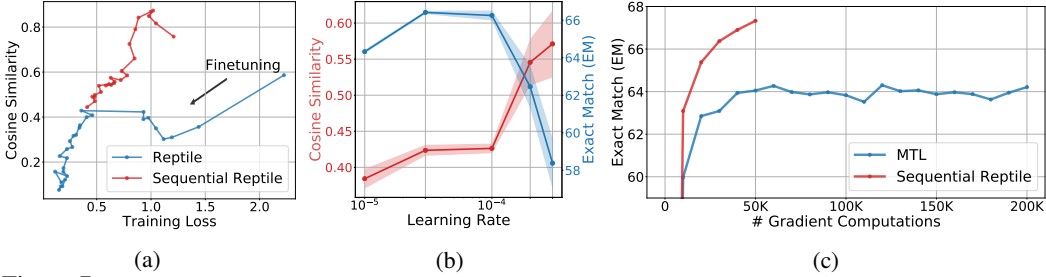

Figure 7: **(a) Trade-off shown in Fig. 1:** average cosine similarity between task gradients vs. MTL training loss. **(b) Effect of the strength of gradient alignment:** Average cosine similarity between task gradients and test performance (EM) vs. inner-learning rate. **(c) Computational efficiency:** Test performance (EM) vs. the cumulative count of (inner-) gradient steps used for training.

task-specific knowledge incompatible across the languages and thereby prevent negative transfer. As a result, Fig. 6a and 6b show that our method can better retain the linguistic knowledge obtained from the pretraining in terms of relatively lower MLM loss. Further, Fig. 6c and 6d confirm this tendency with shorter $\ell_2$ distance from the initially pretrained BERT model.

### 4.4 FURTHER ANALYSIS

**(a) Trade-off shown in Fig. 1:** In Fig. 7a, MTL loss and cosine similarity decrease as we finetune the model from top-right to the bottom-left corner. Meanwhile, Sequential Reptile shows much higher cosine similarities between task gradients at the points of similar MTL losses, achieving better trade-off than Reptile. It explains why simple early stopping cannot outperform Sequential Reptile (see Fig. 7c) that directly enforces gradient alignment across tasks.

**(b) Effect of the strength of gradient alignment:** Then the next question is, how can we further control the trade-off to maximize performance? According to Eq. 7, we can strengthen or weaken the gradient alignment by increasing or decreasing the inner-learning rate $\alpha$, respectively. Fig. 7b shows that while we can control the cosine similarity by varying $\alpha$ as expected, the best-performing $\alpha$ is around $3 \cdot 10^{-5}$, which is indeed the most commonly used value for finetuning the BERT model.

**(c) Computational efficiency:** Lastly, one may suspect training Sequential Reptile takes significantly longer wall clock time because inner steps are not parallelizable as in Reptile (See Fig. 2). This is not true. Fig. 7c shows that whereas base MTL requires around $40K$ gradient computations to achieve 64 EM score, Sequential Reptile requires only around $15K$. As a result, although we run Sequential Reptile with a single GPU at a time, the wall-clock time becomes even comparable to the base MTL that we run in parallel with $8$ GPUs. Please see wall clock comparison on Appendix D.

## 5 CONCLUSION

We showed that when finetuning a well-pretrained language model, it is important to align gradients between the given set of downstream tasks to prevent negative transfer and retain linguistic knowledge acquired from the pretraining. We proposed a simple yet effective method aligning gradients between tasks with efficiency. Specifically, instead of performing multiple inner-optimizations separately for each task, we performed a single inner-optimization by sequentially sampling batches from all the tasks, followed by a Reptile outer update. We extensively validated the efficacy of our method on various MTL and zero-shot cross-lingual transfer tasks, where ours largely outperformed all the baselines we considered.

**Acknowledgements** This work was supported by Institute of Information & communications Technology Planning & Evaluation (IITP) grant funded by the Korea government(MSIT) (No.2019-0-00075, Artificial Intelligence Graduate School Program(KAIST)), the Engineering Research Center Program through the National Research Foundation of Korea (NRF) funded by the Korean Government MSIT (NRF-2018R1A5A1059921), Samsung Research Funding Center of Samsung Electronics (No. SRFC-IT1502-51), Samsung Electronics (IO201214-08145-01), Institute of Information & communications Technology Planning & Evaluation (IITP) grant funded by the Korea government(MSIT) (No. 2021-0-02068, Artificial Intelligence Innovation Hub), the National Research Foundation of Korea (NRF) funded by the Ministry of Education (NRF-2021R1F1A1061655), and KAIST-NAVER Hypercreative AI Center.

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

## A ALGORITHM

We provide the pseudocode for Sequential Reptile described in section 3.1:

---

**Algorithm 1** Sequential Reptile

---

1: **Input:** pretrained language model parameter $\phi$, a set of task-specific data $\{\mathcal{D}_1, \ldots, \mathcal{D}_T\}$, probability vector $(p_1, \ldots, p_T)$ for categorical distribution, the number of inner steps $K$, inner step-size $\alpha$, outer-step size $\eta$.
2: **while** not converged **do**
3:     $\theta^{(0)} \leftarrow \phi$
4:     **for** $k = 1$ to $k = K$ **do**
5:        Sample a task $t_k \sim \text{Cat}(p_1, \ldots, p_T)$
6:        Sample a mini-batch $\mathcal{B}_{t_k}^{(k)}$ from the dataset $D_{t_k}$
7:        $\theta^{(k)} \leftarrow \theta^{(k-1)} - \alpha \dfrac{\partial \mathcal{L}(\theta^{(k-1)}; \mathcal{B}_{t_k}^{(k)})}{\partial \theta^{(k-1)}}$
8:     **end for**
9:     $\text{MG}(\phi) = \phi - \theta^{(K)}$
10:     $\phi \leftarrow \phi - \eta \cdot \text{MG}(\phi)$
11: **end while**

---

Table 6: The number of train/validation instances for each language from TYDI-QA dataset.

| Split | ar | bn | en | fi | id | ko | ru | sw | te | Total |
|-------|------|------|------|------|------|------|------|------|------|--------|
| Train | 14,805 | 2,390 | 3,696 | 6,855 | 5,702 | 1,625 | 6490 | 2,755 | 5,563 | 49,881 |
| Val. | 1,034 | 153 | 449 | 819 | 587 | 306 | 914 | 504 | 828 | 5,594 |

Table 7: The number of train/validation instances for each language from WikiAhn dataset.

| Split | de | en | es | hi | jv | kk | mr | my | sw | te | tl | yo | Total |
|-------|------|------|------|------|------|------|------|------|------|------|------|------|--------|
| Train | 20,000 | 20,017 | 20,000 | 5,001 | 100 | 1,000 | 5,000 | 106 | 1,000 | 10,000 | 100 | 100 | 82,424 |
| Val. | 10,000 | 10,003 | 10,000 | 1,000 | 100 | 1,000 | 1,000 | 113 | 1,000 | 1,000 | 1,000 | 100 | 36,316 |

## B DATASET

**TYDI-QA (Clark et al., 2020)** It is multilingual question answering (QA) dataset covering 11 languages, where a model retrieves a passage that contains answer to the given question and find start and end position of the answer in the passage. Since we focus on extractive QA tasks, we use "Gold Passage" [1] in which ground truth paragraph containing answer to the question is provided. Since some of existing tools break due to the lack of white spaces for Thai and Japanese, the creators of the dataset does not provide the Gold Passage of those two languages. Following the conventional preprocessing for QA, we concatenate a question and paragraph and tokenize it with BertTokenizer (Devlin et al., 2019) which is implemented in transformers library (Wolf et al., 2020). We split the tokenized sentences into chunks with overlapping words. In Table 6, we provide the number of preprocessed training and validation instances for each language.

**WikiAhn (Pan et al., 2017)** It a multilingual NER dataset which is automatically constructed from Wikipedia. We provide the number of instances for training and validation in Table 7.

**XNLI (Conneau et al., 2018)** It is the dataset for multilingual NLI task, which consists of 14 languages other than English. Since it targets for zero-shot cross-lingual transfer, there is no training instances. It provides 7,500 human annotated instances for each validation and test set.

---

[1]https://github.com/google-research-datasets/tydiqa/blob/master/gold_passage_baseline/README.md

Table 8: The number of train/validation instances for each language from MLQA dataset.

| Split | ar | de | en | es | hi | vi | zh | Total |
|-------|-----|-----|--------|-------|-------|-------|-------|--------|
| Val. | 517 | 512 | 1,148 | 500 | 507 | 511 | 504 | 4,199 |
| Test | 5,335 | 4,517 | 11,590 | 5,253 | 4,918 | 5,495 | 5,137 | 42,245 |

**MLQA (Lewis et al., 2020b)**   It is the dataset for zero-shot multilingual question answering task. As XNLI dataset, it only provides only validation and test set for 6 languages other than English. In Table 8, we provide data statistics borrowed from the original paper (Lewis et al., 2020b)

**Common Crawl 100 (Conneau et al., 2020; Wenzek et al., 2020)**   It is multilingual corpus consisting of more than 100 language that is used for pretraining XLM-R (Conneau et al., 2020) model. We download the preprocessed corpus [2] provided by Wenzek et al. (2020). We sample 5,000 instances for each language and evaluate masked language modeling loss.

## C   IMPLICIT GRADIENT ALIGNMENT OF SEQUENTIAL REPTILE

In this section, we provide derivation of implicit gradient alignment of Sequential Reptile in the equation 7. Firstly, we define the following terms from Nichol et al. (2018).

$$\theta^{(0)} = \phi \qquad \text{(initial point)} \tag{8}$$

$$g_{t_k} = \frac{\partial \mathcal{L}(\theta^{(k-1)}; \mathcal{B}_{t_k}^{(k)})}{\partial \theta^{(k-1)}} \quad \text{(gradient obtained during SGD with mini-batch } \mathcal{B}_{t_k}^{(k)}) \tag{9}$$

$$\theta^{(k)} = \theta^{(k-1)} - \alpha g_{t_k} \quad \text{(sequence of parameter vectors)} \tag{10}$$

$$\overline{g}_{t_k} = \frac{\partial \mathcal{L}(\theta^{(k-1)}; \mathcal{B}_{t_k}^{(k)})}{\partial \phi} \quad \text{(gradient at initial point with mini-batch with } \mathcal{B}_{t_k}^{(k)}) \tag{11}$$

$$\overline{H}_{t_k} = \frac{\partial^2 \mathcal{L}(\theta^{(k-1)}; \mathcal{B}_{t_k}^{(k)})}{\partial \phi^2} \quad \text{(Hessian at initial point)} \tag{12}$$

where $\alpha$ denotes a learning rate for inner optimization. $t_k \in \{1, \ldots, T\}$ is a task index and $\mathcal{B}_{t_k}^{(k)}$ is a mini-batch sampled from the task $t_k$.

**Corollary.** *After $K$ steps of inner-updates, expectation of $\sum_{k=1}^{K} g_{t_k}$ over the random sampling of tasks approximates the following:*

$$\mathbb{E}\left[\sum_{k=1}^{K} g_{t_k}\right] \approx \sum_{k=1}^{K} \overline{g}_{t_k} - \frac{\alpha}{2} \sum_{k=1}^{K} \sum_{j=1}^{k-1} \left\langle \overline{g}_{t_k}, \overline{g}_{t_j} \right\rangle \tag{13}$$

where $\langle \cdot, \cdot \rangle$ denotes a dot-product in $\mathbb{R}^d$.

*Proof.* We borrow the key idea from the theorem of Reptile (Nichol et al., 2018) that task specific inner-optimization of Reptile implicitly maximizes inner products between mini-batch gradients within a task. First, we approximate the gradient $g_{t_k}$ as follows.

$$g_{t_k} = \frac{\partial \mathcal{L}(\phi; \mathcal{B}_{t_k}^{(k)})}{\partial \phi} + \frac{\partial^2 \mathcal{L}(\phi; \mathcal{B}_{t_k}^{(k)})}{\partial \phi^2}(\theta^{(k)} - \phi) + O(\|\theta^{(k)} - \phi\|_2^2)$$

$$= \overline{g}_{t_k} - \alpha \overline{H}_{t_k} \sum_{j=1}^{k-1} g_{t_j} + O(\alpha^2)$$

$$= \overline{g}_{t_k} - \alpha \overline{H}_{t_k} \sum_{j=1}^{k-1} \overline{g}_{t_j} + O(\alpha^2)$$

---

[2]http://data.statmt.org/cc-100/

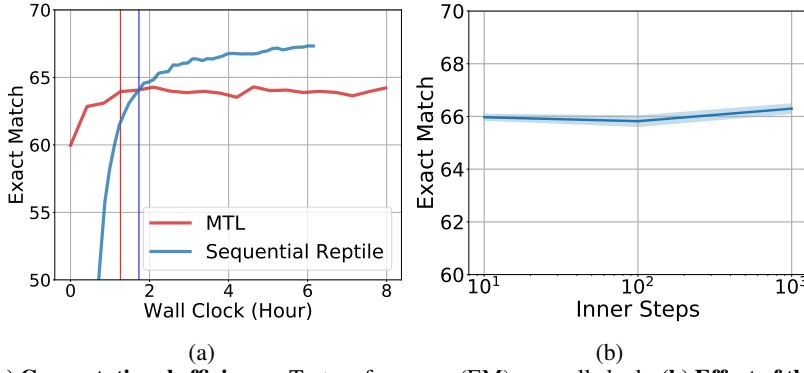

(a)                (b)

Figure 8: **(a) Computational efficiency:** Test performance (EM) vs. wall clock. **(b) Effect of the the number of inner steps:** Test performance (EM) vs the number of inner steps for Sequential Reptile.

After $K$ gradient steps, we approximate the expectation of the meta-gradient $\mathrm{MG}(\phi)$ over the task $t_1, \ldots, t_K$, where $t_k \sim \mathrm{Cat}(p_1, \ldots, p_T)$ for $k = 1, \ldots, K$.

$$
\mathbb{E}\left[\mathrm{MG}(\phi)\right] = \mathbb{E}\left[\sum_{k=1}^{K} g_{t_k}\right] \approx \mathbb{E}\left[\sum_{k=1}^{K} \overline{g}_{t_k} - \alpha \sum_{k=1}^{K}\sum_{j=1}^{k-1} \overline{H}_{t_k}\overline{g}_{t_j}\right]
$$

$$
= \mathbb{E}\left[\sum_{k=1}^{K} \overline{g}_{t_k}\right] - \mathbb{E}\left[\frac{\alpha}{2}\left(\sum_{k=1}^{K}\sum_{j=1}^{k-1} \overline{H}_{t_k}\overline{g}_{t_j} + \sum_{k=1}^{K}\sum_{j=1}^{k-1} \overline{H}_{t_j}\overline{g}_{t_k}\right)\right]
$$

$$
= \mathbb{E}\left[\sum_{k=1}^{K} \overline{g}_{t_k}\right] - \mathbb{E}\left[\frac{\alpha}{2}\sum_{k=1}^{K}\sum_{j=1}^{k-1} \frac{\partial \left\langle g_{t_k}, g_{t_j}\right\rangle}{\partial \phi}\right]
$$

$$
= \mathbb{E}\left[\sum_{k=1}^{K} \overline{g}_{t_k} - \frac{\alpha}{2}\sum_{k=1}^{K}\sum_{j=1}^{k-1} \frac{\partial \left\langle g_{t_k}, g_{t_j}\right\rangle}{\partial \phi}\right]
$$

$$
= \mathbb{E}\left[\sum_{k=1}^{K} \overline{g}_{t_k} - \frac{\alpha}{2}\frac{\partial}{\partial \phi}\left(\sum_{k=1}^{K}\sum_{j=1}^{k-1} \left\langle g_{t_k}, g_{t_j}\right\rangle\right)\right]
$$

$$
= \mathbb{E}\left[\frac{\partial}{\partial \phi}\left(\sum_{k=1}^{K} \mathcal{L}(\phi; \mathcal{B}_{t_k}^{(k)}) - \frac{\alpha}{2}\sum_{k=1}^{K}\sum_{j=1}^{k-1} \left\langle \frac{\partial \mathcal{L}(\phi; \mathcal{B}_{t_k}^{(k)})}{\partial \phi}, \frac{\partial \mathcal{L}(\phi; \mathcal{B}_{t_j}^{(j)})}{\partial \phi}\right\rangle\right)\right]
$$

$$
= \frac{\partial}{\partial \phi}\mathbb{E}\left[\sum_{k=1}^{K} \mathcal{L}(\phi; \mathcal{B}_{t_k}^{(k)}) - \frac{\alpha}{2}\sum_{k=1}^{K}\sum_{j=1}^{k-1} \left\langle \frac{\partial \mathcal{L}(\phi; \mathcal{B}_{t_k}^{(k)})}{\partial \phi}, \frac{\partial \mathcal{L}(\phi; \mathcal{B}_{t_j}^{(j)})}{\partial \phi}\right\rangle\right]
$$

$\square$

Similarly, Smith et al. (2021) show implicit regularization of SGD. After one epoch of SGD, it implicitly biases a model to minimize difference between gradient of each example and full batch gradient. As a result, it approximately aligns gradient of each instance with the full-batch gradient.

## D FURTHER ANALYSIS

**Computational efficiency** In Figure 8a, we plot test Exact Match score as a function of wall clock time. Although training Sequential Reptile is not parallelizable, it shows tolerable computational efficiency compared to MTL model. For MTL, it takes about 1 hour and 15 minutes while Sequential Reptile takes 1 hour and 45 minutes to reach 64 EM score.

**Inner steps** As shown in Figure 8b, Sequential Reptile is robust to the number of steps for the inner optimization. It shows consistent performance with little variance.

Table 9: We train MTL models on 8 tasks from GLUE dataset and report their performance.

| | | | | GLUE | | | | | |
|---|---|---|---|---|---|---|---|---|---|
| **Method** | **CoLA** | **MNLI** | **MRPC** | **QNLI** | **QQP** | **RTE** | **SST2** | **STSB** | **Avg.** |
| **MTL** | 80.82 | **83.47** | 82.10 | **90.50** | 90.16 | 68.23 | 90.13 | **89.73** | 84.39 |
| **RecAdam** | 81.78 | 82.88 | 79.16 | 90.00 | **90.35** | 71.84 | 91.20 | 89.12 | 84.54 |
| **Reptile** | 82.07 | 78.81 | 81.30 | 89.29 | 87.57 | 72.56 | 88.76 | 88.55 | 83.61 |
| **Seq. Reptile** | **82.35** | 83.02 | **83.30** | **90.50** | 89.40 | **73.28** | **92.31** | 89.40 | **85.44** |

Table 10: We finetune MTL models with pretrained ResNet18 backbone on 8 image classification tasks.

| | | | Image Classification | | | | | | |
|---|---|---|---|---|---|---|---|---|---|
| **Method** | **TIN-1** | **TIN-2** | **CIFAR100** | **Dogs** | **Aircraft** | **CUB** | **F-MNIST** | **SVHN** | **Avg.** |
| **MTL** | 56.26 | 53.40 | 54.43 | 33.44 | 44.97 | 29.48 | **90.10** | 87.70 | 56.22 |
| **Reptile** | 23.56 | 23.28 | 9.64 | 12.20 | 10.02 | 6.61 | 60.87 | 20.86 | 20.88 |
| **Seq. Reptile** | **58.12** | **56.83** | **57.46** | **38.05** | **59.55** | **35.96** | 87.44 | **88.86** | **60.28** |

# E ADDITIONAL EXPERIMENTS

In order to show that our model Sequential Reptile is generally applicable to various multi-task learning problems, we additionally perform experiments on monolingual text classification and image classification.

**Text classification** Following (Pilault et al., 2021), we train BERT base model on 7 text classification tasks — CoLA, MNLI, MRPC, QNLI,QQP, RTE, SST2 and one text similarity score regression — STSB from GLUE dataset (Wang et al., 2019a). We share the BERT encoder across all the task and add linear layer on top of the encoder for each task. For Reptile, we use learned initialization with each task specific head for prediction at test time. For STSB task, we use Pearson correlation coefficient to measure the performance of the baselines and ours. For the other 7 task, we evaluate all the models with accuracy. As shown in Table 9, Sequential Reptile outperforms the Reptile and MTL on 4 tasks – CoLA, MRPC, RTE, SST2 and shows comparable performance on the other tasks.

**Image classification** Following the experimental setup from Shin et al. (2021), we use 7 datasets — Tiny-ImageNet (Le & Yang, 2015), CIFAR100 (Krizhevsky et al., 2009), Stanford Dogs (Khosla et al., 2011), Aircraft (Maji et al., 2013), CUB (Wah et al., 2011), Fashion-MNIST (Xiao et al., 2017), and SVHN (Netzer et al., 2011). We class-wisely divide Tiny-ImageNet into two splits which are denoted as TIN-1 and TIN-2, respectively, and consider each split as a distinct task, which results in total 8 tasks for multi-task learning image classification. We finetune ResNet18 (He et al., 2016) which is pretrained on ImageNet (Deng et al., 2009) with a randomly initialized linear classifier for each task. For all the models, the pretrained ResNet is shared across tasks. As the previous experiments, we use shared initialization of Reptile with task specific linear classifiers. We measure accuracy of each image classification task and report the average score.

As shown in Table 10, Sequential Reptile outperforms MTL and Reptile with large margin other than Fashion MNIST (F-MNIST). We observe that Reptile fails on image classification, which is contrast to text classification tasks where it shows reasonable accuracy compared to MTL model. We conjecture that Reptile needs adaptation to the target task since the set of image datasets are more heterogeneous than the set of text datasets.

# F VISUALIZATION OF LEARNING TRAJECTORY

In Figure 9, 10, and 11, we visualize learning trajectory of the MTL models and cosine similarity between task gradients with three different initialization as described in section 4.1. The similar pattern holds for all different initialization. All the MTL baselines except Reptile fall into one of local optima. On the other hand, Sequential Reptile avoid such local minima while maximizing cosine similarities of task gradients.

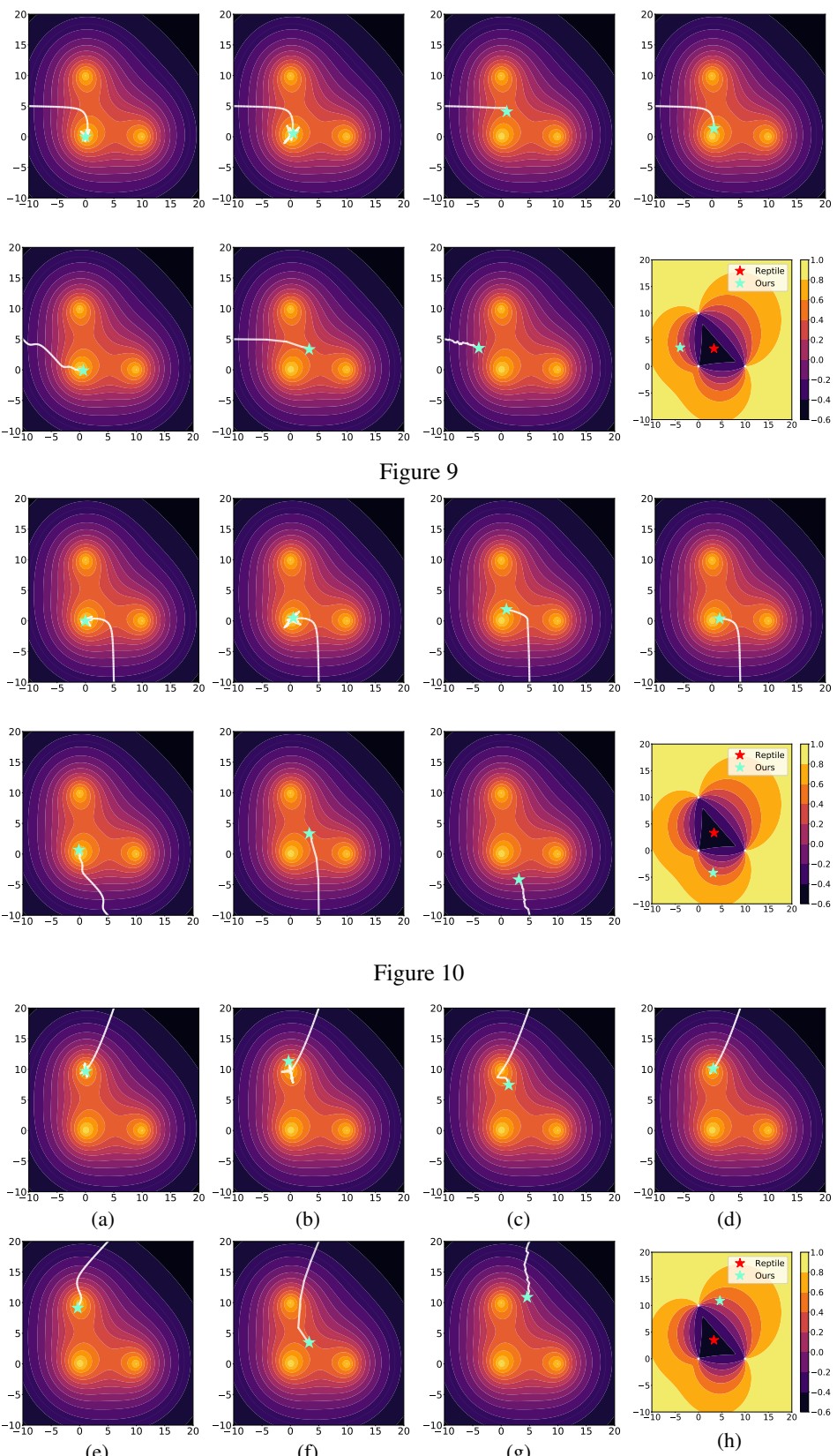

Figure 11: **(a)**∼**(g)** Loss surface and learning trajectory of each method. **(h)** Heatmap shows average pair-wise cosine similarity between the task gradients.

