# OpenReview forum: "Sequential Reptile: Inter-Task Gradient Alignment for Multilingual Learning"
_ICLR.cc/2022/Conference — ICLR 2022 Poster_

### Official Review · Reviewer_8YbF · 2021-11-01

**Correctness:** 3
**Technical Novelty And Significance:** 2
**Empirical Novelty And Significance:** Not applicable
**Recommendation:** 5
**Confidence:** 5

**Main Review:**

Pros:
1. The experimental results are strong and very promising. On all settings considered in this paper, the proposed method significantly outperforms all methods. If this can generalize to other MTL problems, the method has a good potential for the MTL community.
2. The paper provides plenty of experimental evidence to demonstrate the effectiveness of the method. These results also reveal its behavior in the multilingual setting that can be valuable for future research (although I do find some of them to be a bit hard to understand, please see below).

Cons:
1. The novelty of the method is somewhat limited. The proposed modification is just a simple change to the original reptile. In addition, the resulting SR algorithm is actually very similar to a trivial MTL algorithm such that its inner loop is essentially a naive MTL algorithm without within-batch mixing, while the outer loop can be seen as a form of regularization. So indeed the method is a combination of naive MTL with regularization, as suggested in Eq (1). This, however, seems to deviate from the original design principle of Reptile.
2. The intuition behind the method and the source of improvement is not clear to me. This is particularly important given my first point that the method is quite similar to existing methods at the first glance yet shows superiority. There are several points that need better clarification: (1) why are you considering Reptile specifically in the first place? what is the intuition of applying a few-shot learning method here? (2) the intuition behind SR is 'to consider gradient alignment across tasks as well'. While it does sound natural to do so, I find it hard to understand the overall framework with this modification. The original reptile (as well as MAML) aims to find a good initialization for all tasks and therefore considers the bi-level optimization setup which contains task-specific inner loop and task-universal outer loop. Here, this small modification of SR actually fundamentally changes this intuition and set task-universal goals for both inner and outer loops. This does not make sense to me and I wish to get more intuition on this point. (3) The source of improvement is said to be 'our method can effectively filter out language specific knowledge when solving the downstream tasks, which prevents negative transfer and helps retain linguistic knowledge acquired from the pretraining.' But why is that case? The results have shown that the resulting model is closer to the pretrained mBERT. But there is nothing specifically designed for SR to do so. The outer loop can be a potential cause yet regular reptile also has this step (while being much less similar in terms of L2 distance). I think it will be helpful if more analysis can be included to show the source of improvement.

Minor issues/questions/suggestions:
1. It seems to me that adding regularization towards the original mBERT is helpful for the performance. Can we add it for other baselines considered (or perhaps other related techniques introduced in [1,2,3])? Would that improve their performance?
2. How about using within-batch mixing for the proposed method? How would that compare against the current version?

[1] Noise Stability Regularization for Improving BERT Fine-tuning. Hua et al., 2021.
[2] Mixout: Effective Regularization to Finetune Large-scale Pretrained Language Models. Leet et al., ICLR 2020.
[3] SMART: Robust and Efficient Fine-Tuning for Pre-trained Natural Language Models through Principled Regularized Optimization. Jiang et al., ACL 2020.

**Summary Of The Paper:**

The paper presents a new method for multi-task learning (MTL), namely sequential reptile (SR). The proposed method aims to align task gradients and mitigate negative interference among tasks during the finetuning process of multilingual BERT. Specifically, different from a regular Reptile which performs the inner loop on a single (sampled) task, the modified version does so on a batch of tasks, sampled according to some prior distributions. The paper argues that this simple modification can promote gradient alignment naturally. Empirically, the proposed SR method outperforms all baselines considered significantly, while using similar or even less computational resources.

**Summary Of The Review:**

Overall, this paper presents a simple method with strong empirical improvements, yet the intuition and source of improvement can be somewhat confusing.

---

> ### Author Response · Authors · 2021-11-12
> **Repose to reviewer 8YbF (3/3)**
>
> # References
> [1] Nichol, Alex, Joshua Achiam, and John Schulman. "On first-order meta-learning algorithms." arXiv preprint arXiv:1803.02999 (2018).
>
> [2] Flennerhag, Sebastian, et al. "Transferring Knowledge across Learning Processes." International Conference on Learning Representations. 2019.
>
> [3] Shin, Jaewoong, et al. "Large-Scale Meta-Learning with Continual Trajectory Shifting." International Conference on Machine Learning. PMLR, 2021.
>
> [4] Riemer, Matthew, et al. "Learning to Learn without Forgetting by Maximizing Transfer and Minimizing Interference." International Conference on Learning Representations. 2019.
>
> [5] Samuel L Smith, Benoit Dherin, David Barrett, and Soham De.  On the origin of implicit regularization in stochastic gradient descent.  InInternational Conference on Learning Representations,2021.
>
> [6] Hua, Hang, et al. "Noise Stability Regularization for Improving BERT Fine-tuning." Proceedings of the 2021 Conference of the North American Chapter of the Association for Computational Linguistics: Human Language Technologies. 2021.
>
> [7] Lee, Cheolhyoung, Kyunghyun Cho, and Wanmo Kang. "Mixout: Effective Regularization to Finetune Large-scale Pretrained Language Models." International Conference on Learning Representations. 2020.
>
>
> [8] Jiang, Haoming, et al. "SMART: Robust and Efficient Fine-Tuning for Pre-trained Natural Language Models through Principled Regularized Optimization." Proceedings of the 58th Annual Meeting of the Association for Computational Linguistics. 2020.

---

> ### Author Response · Authors · 2021-11-12
> **Repose to reviewer 8YbF (2/3)**
>
> **[Q3]** The intuition behind SR is 'to consider gradient alignment across tasks as well'. While it does sound natural to do so, I find it hard to understand the overall framework with this modification. The original reptile (as well as MAML) aims to find a good initialization for all tasks and therefore considers the bi-level optimization setup which contains task-specific inner loop and task-universal outer loop. Here, this small modification of SR actually fundamentally changes this intuition and set task-universal goals for both inner and outer loops. This does not make sense to me and I wish to get more intuition on this point.
>
> - What you are confused about is precisely our novelty here. As you said, the original Reptile has the **task-specific inner-loop** and the **task-universal outer-loop**, which we also compare in our experiments. However, the limitation of this approach is that **each task-specific gradient has no reason to be well aligned with the gradients of other tasks**. This is because each task-specific optimization is done independently of the other tasks. The outer-gradient of this procedure is shown in Eq.(4) in our paper. Note that the dot product is computed only between the batches sampled from the same task.
>
> - On the contrary, our algorithm impose **strong dependency across batches sampled from the different tasks**. Specifically, we maximize the dot product between them as shown in Eq.(7). This can be done by sequentially sampling batches from different tasks to draw an inner-trajectory (proof in Appendix C). Interestingly, the inner-optimization is no longer “task-specific”, but task-universal as you pointed out. However, this is completely okay for our setting since we are no longer confined within the meta-learning framework, and the learned parameter is no longer a shared initialization. It’s not the case that we are given a novel task and should adapt to it. Rather, according to Eq.(7), we simply want to enforce inter- and intra-task gradient alignment on the **original MTL objective**. In terms of achieving this goal, Sequential Reptile is the straightforward answer. In sum, our sequential reptile is a **MTL algorithm and not a meta-learning algorithm**, and has a **completely different effect** from Reptile (**gradient alignments between the given set of tasks**).
>
> - As for why Eq.(7) holds, we strongly recommend you to read Appendix C or the original Reptile paper section 5.1.
>
> ---
>
> **[Q4]** The source of improvement is said to be 'our method can effectively filter out language specific knowledge when solving the downstream tasks, which prevents negative transfer and helps retain linguistic knowledge acquired from the pretraining.' But why is that case? The results have shown that the resulting model is closer to the pretrained mBERT. But there is nothing specifically designed for SR to do so. The outer loop can be a potential cause yet regular reptile also has this step (while being much less similar in terms of L2 distance). I think it will be helpful if more analysis can be included to show the source of improvement.
>
> - Although Reptile consists of inner- and outer-loop as you mentioned, Reptile **only promotes intra-task gradient alignments** because each **inner-optimization is task-specific (see Eq.(4))**. On the other hand, our Sequential Reptile can promote **inter-task gradient alignments** because all the tasks sequentially alternate within the single inner-optimization, followed by an outer update (see Eq.(7)). Thus the effect of the two are completely different, and the **inter-task gradient alignment** allows the Sequential Reptile to alleviate task-specific overfitting, negative transfer, and catastrophic forgetting of the pretrained knowledge.
>
> ---
> **[Q5]** It seems that adding regularization [6,7,8] to the original mBERT is helpful for the performance. Can we add it for other baselines considered?
>
> - We have already performed experiments by adding regularizations such as GradNorm or RecAdam to mBERT.  However, **they cannot align task gradients (none of the baselines can)** and thus do not prevent negative transfer as described in the paper.  In contrast, Sequential Reptile effectively tackles negative interference thanks to its inter-task gradient alignment.

---

> ### Author Response · Authors · 2021-11-12
> **Repose to reviewer 8YbF (1/3)**
>
> We really appreciate your time and effort for the detailed comments. We respond each of the comments below.
>
> **[Q1]** The novelty of the method is somewhat limited. The proposed modification is just a simple change to the original reptile. In addition, the resulting SR algorithm is actually very similar to a trivial MTL algorithm such that its inner loop is essentially a naive MTL algorithm without within-batch mixing, while the outer loop can be seen as a form of regularization. So indeed the method is a combination of naive MTL with regularization, as suggested in Eq (1). This, however, seems to deviate from the original design principle of Reptile.
>
> - This seems like a misunderstanding. Despite the simplicity of our sequential task sampling algorithm, it was never introduced before, although it could bring in impressive performance gains as we show in our experiments. We also did not only propose the method, but provided many novel insights as to why it improves performance in terms of inter-task gradient alignment, negative transfer, and catastrophic forgetting, which can help further advance the research in multi-task learning.
>
> - Also, the simplicity of the algorithm is not a weakness, but rather its strength, as it is easy to implement. Thus, we believe that it will have a large practical impact, and will serve as a useful baseline for future research on the topic.
>
> - The underlying mechanism also follows the original design principle of Reptile as well. Please refer to the original Reptile paper (section 5.1), where the authors analyzed the effect of intra-task gradient-alignment with Taylor approximation. Our intuition and justification shares the same principle, except that we extend it to inter-task alignment.
>
> ---
>
> **[Q2]** Why are you considering Reptile specifically in the first place? What is the intuition of applying a few-shot learning method here?
>
> - Reptile [1] has recently been used not only for few-shot learning, but also for many-shot learning [2,3] and continual learning [4] as well. These days Reptile is recognized as a very general learning framework not specifically tailored to few-shot meta-learning. Its connection to standard learning procedures has already been well understood by recent studies [4,5]
>
> - Among many important properties of Reptile, the one that we make use of is as follows, which is closely related to [4,5]:  **We can interpret the Reptile outer update as a regularized version of (sum of) inner-updates (See Eq.(4)).** See Appendix C for the actual derivation. Suppose we have a standard learning problem on which we want to impose a gradient-alignment regularization. Then, according to the Reptile property above, we can impose the regularization on this problem simply by forming inner- and outer- loops as with Reptile.
>
> - In doing so, we did not introduce **any notion of meta-learning or shared initialization**. Since we are not given a novel task at test time, there is no need for further adaptation. The learned parameters do not have to be a shared initialization, either. This interpretation explains how Reptile is connected to other standard learning problems such as continual learning [4], or even SGD procedure itself [5].

---

> ### Author Response · Authors · 2021-11-25
> **A Gentle Reminder**
>
> Dear reviewer
>
> We sincerely appreciate your efforts in reviewing our paper, and your constructive comments. We have responded to your comments, faithfully reflected them in the revision, and provided additional experimental results that you have requested. Could you please go over our responses and the revision since end of the final discussion phase is approaching? Please let us know there is anything else we need to clarify or provide.
>
> Thanks, authors.

---

### Official Review · Reviewer_gNwa · 2021-11-03

**Correctness:** 3
**Technical Novelty And Significance:** 2
**Empirical Novelty And Significance:** 2
**Recommendation:** 5
**Confidence:** 4

**Main Review:**

Strength:
- This paper tackles a problem which is less understood nor solved. The experiments results demonstrate the empirical significance of the work.
- The method is simple and the authors provide insights into the problem and the methods, i.e. the adaptation of Reptile.
- The experiments design are reasonable, specifically the zero-shot crosslingual transfer experiments are interesting.
- The overall writing is clear and the content flows well.

Weakness:
- One major drawback of the paper is that it mixes two problems, i.e. catastrophic forgetting (of knowledge in pretrained model) vs. negative transfer between tasks (languages) during finetuning. It seems the authors focus on solving the second problem, but they also hypothesize that solving the second problem would improve the first problem as a by-product. For example, the paper made several claims such as "As a result of such negative transfer, we see from Fig. 5a and Fig. 5b that the baselines suffer from catastrophic forgetting.". However, such hypothesis (that the former caused the latter) was never verified.
- Another limitation is the incompleteness in experimentation. For example, the authors use BERT base, which is a relative small model and draw conclusions such as "It implies that the baselines suffer from negative transfer while ours is relatively less vulnerable.". Several work has shown that negative transfer is related to model capacity, e.g. capacity bottleneck has been well studied in MTL literature and specifically in multilingual translation [1]. Therefore, I'd like to see experiments with larger models to verify that this method is still as effective in mitigating negative transfer as was observed in small models.
- Related to the above point, it's unclear whether some of the worse performance from baselines are due to specific experiment setup which puts those approaches at disadvantage. For example, the authors shows that "We see that all the baselines butours highly degrade the performance on high resource languages" but the experiment was conducted in a manner where such observation may be expected, i.e. the authors adjust the sampling distribution to pt∝(Nt)^1/5, which upsamples low resource while downssamples high resource, that is, it intrinsically hurts high resource languages' performance.


The technical details in several places are not clear or lack of justification:
1. In Eq. 1, the MTL loss is defined as the sum of single task losses. This definition is quite limited since minimizing this loss may not be the objective of multi-task learning. A more general MTL objective has been studied as a Pareto front of task losses [2].
2. This work points out the importance of optimization setting which leads to different learning trajectories. However, the experiments were done with a specific set of hyperparameter, e.g. the authors chose 8 tasks (languages) in inner loop and sampled them with a specific distribution, the batch sizes were chosen differently for different tasks (e.g. QA vs. NLI). However, the author did not provide empirical justification that how senstive is the method to the choice of those hyperparameters.
3. In Fig 5.c, Why RecAdam has the largest L2 distance given that it has explicit objective to regularize this metric?
4. The clustering preprocessing in zero-shot crosslingual transfer experiments are very unclear. Why do you need to do the clustering to form tasks instead of using language as tasks as is in other experiments?
5. Fig 7.c does not exactly capture the training overhead from the proposed method. Could you provide a wall clock based comparison (which is more precise) as you mentioned in the text?
6. Related work misses relevant work on the optimization methods of multilingual training, e.g. [3] and [4].

[1] Lepikhin, Dmitry, et al. "GShard: Scaling Giant Models with Conditional Computation and Automatic Sharding." International Conference on Learning Representations. 2020.
[2] Lin, Xi, et al. "Pareto multi-task learning." Advances in neural information processing systems 32 (2019): 12060-12070.
[3] Wang, Xinyi, Yulia Tsvetkov, and Graham Neubig. "Balancing Training for Multilingual Neural Machine Translation." Proceedings of the 58th Annual Meeting of the Association for Computational Linguistics. 2020.
[4] Li, Xian, and Hongyu Gong. "Robust Optimization for Multilingual Translation with Imbalanced Data." arXiv preprint arXiv:2104.07639 (2021).

**Summary Of The Paper:**

This paper proposed a new training method to improve gradients alignment between languages (tasks) in multilingual finetuning of pretrained language models. The method is based on an existing algorithm, Reptile, which was originally developed for meta learning. The authors adapted it for multi-task learning and verified its effectiveness in multilingual finetuning.

**Summary Of The Review:**

This paper proposes a simple method to improve multilingual finetuning of pretrained language models. The proposed solution and its effectiveness indicated by the experiments results has potential for enough technical significance. However, the current version of the paper has room for improvement in terms of experiment completeness in order to draw conclusion on the practical value of the proposed method.

---

> ### Author Response · Authors · 2021-11-12
> **Response to reviewer gNwa (3/3)**
>
> # References
> [1] Lepikhin, Dmitry, et al. "GShard: Scaling Giant Models with Conditional Computation and Automatic Sharding." International Conference on Learning Representations. 2020.
>
> [2] Lin, Xi, et al. "Pareto multi-task learning." Advances in neural information processing systems 32 (2019): 12060-12070.
>
> [3] Wang, Xinyi, Yulia Tsvetkov, and Graham Neubig. "Balancing Training for Multilingual Neural Machine Translation." Proceedings of the 58th Annual Meeting of the Association for Computational Linguistics. 2020.
>
> [4] Li, Xian, and Hongyu Gong. "Robust Optimization for Multilingual Translation with Imbalanced Data." arXiv preprint arXiv:2104.07639 (2021).
>
> [5] Carpenter, Gail A., and Stephen Grossberg. "A massively parallel architecture for a self-organizing neural pattern recognition machine." Computer vision, graphics, and image processing 37.1 (1987): 54-115.
>
>
> [6] Riemer, Matthew, et al. "Learning to Learn without Forgetting by Maximizing Transfer and Minimizing Interference." International Conference on Learning Representations. 2019.
>
> [7] Look-ahead meta learning for continual learning. Advances in Neural Information Processing Systems , 33, 2020.
>
> [8] Sun, Zhiqing, et al. "MobileBERT: a Compact Task-Agnostic BERT for Resource-Limited Devices." Proceedings of the 58th Annual Meeting of the Association for Computational Linguistics. 2020.
>
> [9] Wang, Wenhui, et al. "Minilm: Deep self-attention distillation for task-agnostic compression of pre-trained transformers."  Advances in Neural Information Processing Systems , 33, 2020.
>
> [10] Arivazhagan, Naveen, et al. "Massively multilingual neural machine translation in the wild: Findings and challenges." arXiv preprint arXiv:1907.05019 (2019).
>
> [11] Wang, Zirui, Zachary C. Lipton, and Yulia Tsvetkov. "On Negative Interference in Multilingual Language Models." Proceedings of the 2020 Conference on Empirical Methods in Natural Language Processing (EMNLP). 2020.
>
>
> [12] Wang, Zirui, et al. "Gradient Vaccine: Investigating and Improving Multi-task Optimization in Massively Multilingual Models." International Conference on Learning Representations. 2021.
>
> [13] Hsu, Kyle, Sergey Levine, and Chelsea Finn. "Unsupervised Learning via Meta-Learning." International Conference on Learning Representations. 2019.
>
> [14] Aharoni, Roee, and Yoav Goldberg. "Unsupervised Domain Clusters in Pretrained Language Models." Proceedings of the 58th Annual Meeting of the Association for Computational Linguistics. 2020.

---

> ### Author Response · Authors · 2021-11-12
> **Response to reviewer gNwa (2/3)**
>
>
> **[Q4]** In Eq. 1, the MTL loss is defined as the sum of single task losses. This definition is quite limited since minimizing this loss may not be the objective of multi-task learning. A more general MTL objective has been studied as a Pareto front of task losses [2].
>
> - Eq.(1), the sum of single task losses, is the standard multi-task learning objective researchers have investigated for a long time. Although there has been some efforts in better weighing the task-specific losses for MTL, most of the MTL methods are based on the Eq.(1). Also, our method is an optimization algorithm and how to optimally combine the task loss for MTL is **completely orthogonal** to our method.
>
> ---
> **[Q5]** This work points out the importance of optimization setting which leads to different learning trajectories. However, the experiments were done with a specific set of hyperparameters, e.g. the authors chose 8 tasks (languages) in an inner loop and sampled them with a specific distribution, the batch sizes were chosen differently for different tasks (e.g. QA vs. NLI). However, the author did not provide empirical justification for how sensitive the method is to the choice of those hyperparameters.
>
> - We sampled 8 tasks since we have 8 gpus in a single server for parallelization. The specific sampling distribution is borrowed from the existing works [11,12,13]. Batch sizes were chosen based on GPU memory (2080 RTX-ti). Those **hyperparameters are not specific to our model, and the same hyperparameters were used for all baselines**. We select them based on convention and our computational budget. Therefore, there is no particular reason to show that our method is robust against them.
>
> - Rather, the inner-learning rate is the most relevant hyperparameter to our model, and we already show in Figure 7(b) that it is important to carefully select the inner-learning rate to find a good tradeoff between minimizing MTL loss and maximizing cosine similarity.
> - Additionally, we have included Figure 8(b) which shows test EM performance as a function of the number of inner steps. Sequential Reptile shows consistent performance with little variance.
>
> ---
>
> **[Q6]** In Fig 5.c, Why does RecAdam show the largest L2 distance given that its objective explicitly regularizes this metric?
> - We use the **official code** (https://github.com/Sanyuan-Chen/RecAdam ) from the authors  and use the **same hyperparameters** for controlling the scheduling and intensity of the regularization. We conjecture that RecAdam shows large L2 distance because it **gradually decreases the strength of L2-regularization** as training goes on, ending up with the vanilla MTL objective.
> - Moreover, we have tried slightly increasing the intensity of L2 regularization for RecAdam. Even though such stronger regularization successfully minimizes L2 distance from the initial pretrained model, **it fails to minimize the MTL loss**. Thus, it seems difficult to stabilize the training if we increase the regularization strength.
> ---
> **[Q7]** The reason why clustering is necessary for zero-shot cross-lingual transfer experiments is not clear.
>
> - It is already **explained in page 8, in the “Implementation Detail” paragraph**. Since we are given a single language (English) for zero-shot cross-lingual transfer, we need to construct multiple tasks from a single language. Therefore, following [14,15], we cluster the training data into four groups and consider each group as a distinct task.
> ---
>
> **[Q8]** Could you provide a wall clock based comparison to capture the training overhead from the proposed method?
> -  We included the figure which plots EM score vs. wall clock time in **Appendix D**. As mentioned in the paper, Sequential Reptile **trained on a single GPU** only requires roughly **1.4x wall clock time** to achieve 64 EM score, compared to the **base MTL trained on 8 GPUs**.
>
> ---
>
> **[Q9]** Related work misses relevant work on the optimization methods of multilingual training, e.g. [3] and [4].
> - Please note that we have **already cited [3]** in the related work section. We have **included [4]** in the revision as suggested.

---

> ### Author Response · Authors · 2021-11-12
> **Response to reviewer gNwa (1/3)**
>
> We sincerely appreciate your constructive comments. We respond to the individual comments below:
>
> **[Q1]** One major drawback of the paper is that it mixes two problems, i.e. catastrophic forgetting (of knowledge in a pretrained model) vs. negative transfer between tasks (languages) during finetuning. It seems the authors focus on solving the second problem, but they also hypothesize that solving the second problem would improve the first problem as a by-product. For example, the paper made several claims such as "As a result of such negative transfer, we see from Fig. 5a and Fig. 5b that the baselines suffer from catastrophic forgetting.". However, such a hypothesis (that the former caused the latter) was never verified.
> - In the context of continual learning, it is widely known that maximizing adaptation and minimizing catastrophic forgetting are very closely related, i.e. they are in a trade-off relationship. In this view, catastrophic forgetting is nothing but a phenomenon of negative transfer. In other words, catastrophic forgetting happens when a model fails to stay in balance because of excessive adaptation to the new task. In our paper, we have also discussed this problem in the introduction section (see the second paragraph and Figure 1) and empirically verified in Figure 7(a).
> - Formally speaking, this conceptualization is called stability-plasticity dilemma [5], where stability means being robust against forgetting and plasticity means that the learner should sufficiently adapt. Later, MER [6] and La-MAML [7] show that we can achieve a good tradeoff by making use of gradient alignment. Our work shares the same conceptualization as those literatures. Please refer to them for further justification.
>
> ---
>
> **[Q2]** Another limitation is the incompleteness in experimentation. For example, the authors use BERT base, which is a relatively small model and draw conclusions such as "It implies that the baselines suffer from negative transfer while ours is relatively less vulnerable." Several works have shown that negative transfer is related to model capacity, e.g. capacity bottleneck has been well studied in MTL literature and specifically in multilingual translation [1]. Therefore, I'd like to see experiments with larger models to verify that this method is still as effective in mitigating negative transfer as was observed in small models.
>
> -  First and most importantly, **gradient alignment between tasks with our algorithm promotes positive transfer**, which is equally important even for a larger model that may suffer less from inter-task conflicts.
>
> - Secondly, BERT is one of the most widely used models for real-world NLU applications (https://blog.google/products/search/search-language-understanding-bert/), and we believe that the experimental setup we considered is practical and realistic enough to draw a conclusion on our method's effectiveness. Also, even if large-scale language models such as GPT-3 may alleviate catastrophic forgetting better, due to the computational and memory limitation, we do believe that BERT will remain a more practical choice.
>
> - Also, increasing the model capacity will require a large amount of computation and memory overhead. Why shouldn't we use a method that can solve the problem without sacrificing the efficiency?
>
> - Moreover, training large-scale language models require a considerable amount of computational resources such as TPU, which is not available to all researchers, including ourselves. Also, even if we had sufficient resources to train such a model for once, it is almost impossible to perform extensive experimental validation as we have done in our paper, with such a large language model. For the case of Bert-Large, the multilingual version of it is not even publicly available. Thus, considering a larger language models is impractical.
>
> ---
>
> **[Q3]** Related to the above point, it's unclear whether some of the worse performance from baselines are due to specific experiment setup which puts those approaches at disadvantage. For example, the authors shows that "We see that all the baselines but ours highly degrade the performance on high resource languages" but the experiment was conducted in a manner where such observation may be expected, i.e. the authors adjust the sampling distribution to pt∝(Nt)^1/5, which up samples low resource while down samples high resource, that is, it intrinsically hurts high resource languages' performance.
>
> - This is a **factual misunderstanding**. We explicitly mentioned in our paper (section 4.2 “Implementation Details” paragraph) that for all the baselines and our model, we use **exactly the same sampling distribution $p_t \propto (N_t)^{1/5}$**, which is commonly used in MTL for NLP [11,12,13]. Throughout all our experiments, we carefully tuned all hyperparameters for all baselines, so that there is no fairness issue.

---

> ### Author Response · Authors · 2021-11-25
> **A Gentle Reminder**
>
> Dear reviewer
>
> We sincerely appreciate your efforts in reviewing our paper, and your constructive comments. We have responded to your comments, faithfully reflected them in the revision, and provided additional experimental results that you have requested. Could you please go over our responses and the revision since end of the final discussion phase is approaching? Please let us know there is anything else we need to clarify or provide.
>
> Thanks, authors.

---

### Official Review · Reviewer_vCuD · 2021-11-03

**Correctness:** 3
**Technical Novelty And Significance:** 3
**Empirical Novelty And Significance:** 3
**Recommendation:** 5
**Confidence:** 4

**Main Review:**

Strength

1.The paper is well-organized and clearly presented.

2.The proposed Sequential Reptile is very simple and extensive experiments show that Sequential Reptile can significantly outperform baseline methods.

Weaknesses

1.Negative transfer and catastrophic forgetting are the common issues in the transfer learning field, which are not specific for multilingual learning. Besides, the proposed Sequential Reptile method is also general (simply modify the task sampling strategy to a sequential way during the inner optimization), which also has not special designs for the multilingual setting. I am curious why the authors choose this setting instead of more general transfer learning settings to verify the effectiveness of the proposed method.

2.The authors claim that finetune a model with MTL objective will gradually enforce the model to memorize task-specific knowledge. Did the
authors compare with some transfer learning methods like multi-task adversarial training to learn task-invariant representations?

3.The experiments only focus on two tasks, i.e., QA and NER. It would be better to verify on more multilingual tasks, e.g., XGLUE and Xtreme.

4.Are the improvements statistically significant?


**Summary Of The Paper:**

This paper proposes Sequential Reptile, a meta-learning method to perform inter-task gradient alignment to alleviate negative transfer and catastrophic forgetting. Empirically, experiments on both multi task learning and zero-shot cross-lingual transfer settings for QA and NER tasks demonstrate the effectiveness of the proposed method.

**Summary Of The Review:**

I carefully review this paper. I would vote for marginally below the acceptance threshold and make the final decision after seeing the rebuttal from authors .

---

> ### Author Response · Authors · 2021-11-15
> **Response to Reviewer vCuD (2/2)**
>
>
> **[Q2]** The authors claim that fine-tuning a model with an MTL objective will gradually enforce the model to memorize task-specific knowledge. Did the authors compare with some transfer learning methods like multi-task adversarial training to learn task-invariant representations?
> -  In the revision, we **newly included another baseline [1]**  which learns task invariant representation with adversarial training. **Sequential Reptile still outperforms that baseline for all the settings**. Note that we carefully tuned the coefficient for adversarial training since the model is very sensitive to how it balances between minimizing adversarial loss and MTL loss.
> - Moreover the adversarial training degrades the accuracy on zero-shot cross-lingual transfer. We conjecture this is due to the lack of heterogeneity of source training data since the training data consists of a monolingual training set. It may fail to learn language invariant representations that generalize to unseen languages at test time.
> ---
> **[Q3]** The experiments only focus on two tasks, i.e., QA and NER. It would be better to verify on more multilingual tasks, e.g., XGLUE and Xtreme.
> - Thank you for your suggestion. However, XGLUE [2] and Xtreme [3] are not multilingual tasks since the dataset consists of a monolingual corpus. They are for zero-shot cross-lingual transfer tasks. **We have already performed zero-shot cross-lingual transfer experiments with MLQA and XNLI datasets** in section 4.3, which are **indeed part of XGLUE and Xtreme datasets**. We emphasize that we focus on more than two tasks (**total 6 tasks**: multilingual-QA, multilingual-NER, MLQA, XNLI, GLUE, Image classification).
>
> ---
> **[Q4]** Are the improvements statistically significant?
> -  We have performed **a pairwise t-test** between Sequential Reptile and the other baselines for the QA task. We get p-value less than 0.05 and thus we can conclude **the improvements are statistically significant**.
>
>
> |   Method  | p-value |
> |--------|---------|
> | MTL      | 0.0106  |
> | PCGrad   | 0.0007  |
> | GradVac  | 0.0126  |
> | GradNorm | 0.0105  |
> | RecAdam  | 0.0023  |
> | Reptile  | 0.0126  |
>
> ---
> # References
>
> [1] Seanie Lee, Donggyu Kim, and Jangwon Park. "Domain-agnostic Question-Answering with Adversarial Training." EMNLP 2019 MRQA Workshop. 2019.
>
> [2] Liang, Yaobo, et al. "Xglue: A new benchmark dataset for cross-lingual pre-training, understanding and generation." arXiv preprint arXiv:2004.01401 (2020).
>
> [3] Hu, Junjie, et al. "Xtreme: A massively multilingual multi-task benchmark for evaluating cross-lingual generalisation." International Conference on Machine Learning. PMLR, 2020.

---

> ### Author Response · Authors · 2021-11-15
> **Response to Reviewer vCuD (1/2)**
>
> We sincerely appreciate your constructive comments. We respond to the individual comments below:
>
>
> **[Q1]** Negative transfer and catastrophic forgetting are the common issues in the transfer learning field, which are not specific for multilingual learning. Besides, the proposed Sequential Reptile method is also general (simply modify the task sampling strategy to a sequential way during the inner optimization), which also has no special designs for the multilingual setting. I am curious why the authors choose this setting instead of more general transfer learning settings to verify the effectiveness of the proposed method.
> - Thank you for your insightful comment. Following your suggestion, **we have conducted MTL experiments on the GLUE dataset as well as multi-task image classifications**. They are different from the multilingual learning task in that we need a separate task-specific head (classifier or regressor) for each task.
> - Again, **Sequential Reptile largely outperforms the relevant baselines**, demonstrating its effectiveness for solving general MTL problems. We have included the new experimental results in the revision (Appendix E).
> ---
> **< General MTL setting 1: GLUE >**
>
> | **Method**       |    **CoLA**   |    **MNLI**   |    **MRPC**   |    **QNLI**   |    **QQP**    |    **RTE**    |    **SST2**   |    **STSB**   |    **Avg.**    |
> |--------------|:---------:|:---------:|:---------:|:---------:|:---------:|:---------:|:---------:|:---------:|:---------:|
> | MTL          |   80.82   | **83.47** |    82.1   | **90.50** | **90.16** |   68.23   |   90.13   | **89.73** |   84.39   |
> | Reptile      |   82.07   |   78.81   |    81.3   |   89.29   |   87.57   |   72.56   |   88.76   |   88.55   |   83.61   |
> | **Seq. Reptile** | **82.35** |   83.02   | **83.30** | **90.50** |    89.4   | **73.28** | **92.31** |    89.4   | **85.44** |
>
> ---
> **< General MTL setting 2: Image Classification >**
>
> | **Method**       | **TIN-1**     | TIN-2     | CIFAR100  | Dogs      | Aircraft  | CUB       | FMNIST    | SVHN      | Avg.      |
> |--------------|:---------:|:---------:|:---------:|:---------:|:---------:|:---------:|:---------:|:---------:|:---------:|
> | MTL          |     56.26 |      53.4 |     54.43 |     33.44 |     44.97 |     29.48 | **90.19** |     87.70 |     56.22 |
> | Reptile      |     23.56 |     23.28 |      9.64 |      12.2 |     10.02 |      6.61 |     60.87 |     20.86 |     20.88 |
> | **Seq. Reptile** | **58.12** | **56.83** | **57.46** | **38.05** | **59.55** | **35.96** |     87.44 | **88.86** | **60.28** |

---

> ### Author Response · Authors · 2021-11-25
> **A Gentle Reminder**
>
> Dear reviewer
>
> We sincerely appreciate your efforts in reviewing our paper, and your constructive comments. We have responded to your comments, faithfully reflected them in the revision, and provided additional experimental results that you have requested. Could you please go over our responses and the revision since end of the final discussion phase is approaching? Please let us know there is anything else we need to clarify or provide.
>
> Thanks, authors.

---

### Official Review · Reviewer_ZkcB · 2021-11-07

**Correctness:** 3
**Technical Novelty And Significance:** 3
**Empirical Novelty And Significance:** 3
**Recommendation:** 6
**Confidence:** 3

**Main Review:**

Strengths:
- The hypothesis is very interesting: is the negative transfer actually caused by the misalignments of task gradients? The paper provides some empirical evidences showing that such correlation (may or may not in causality relationship though) did exist, through a large number of synthetic and multilingual experiments.
- The proposed sequential reptile algorithm is simple and effective.
- The experiments are pretty extensive and show many interesting insights, which also verifies the effectivenesses of the proposed method. I think the following insights can be important:

a). Both reptile and sequential reptile show better generalizability across different tasks of the final optimized model.

b). By implicitly adding the task gradient alignment, the resulted finetuned MTL model show less forgetting of the pretrained knowledge, and higher cosine similarities across tasks.

c) The task gradient alignment does have correlation with the MTL task model performance, though it's still not sure if they are in causality relationship.

Weaknesses:
- The paper's findings would be much more generalizable if the experiments are also performed in the MTL setting of a single language, i.e., in a standard MTL setting.
- There are still a few issues in the experiments:

a) In the synthetic experiment, instead of only showing the loss landscape of a single initialization point, it would be more convincing if such patterns hold for random initialization.

b) In Table 1&2, the average performance is calculated as the mean performance across different languages. Such average doesn't consider the differences of the number of train/eval examples for different languages. Typically, we observed most MTL methods hurt for high-resource languages while improve on low-resource language. Simply averaging across different languages lead to a discrepancy between MTL model training and results reporting, as the loss is optimized as the sum of the example losses in training time. Therefore, simply averaging the performance across languages can be a misleading performance report as the actual empirical effectiveness per example may differ. But I guess this issue exists in most MTL papers.

c) In Table 5 & 6, it would be good to include the single-task learning (STL) performance on zero-shot cross-lingual experiments.

d) More experiments on some hyper-parameters could lead to better understanding of the algorithms, especially on inner and outer learning rates and inner gradient steps.


**Summary Of The Paper:**

The paper proposed an optimization method for training multi-task learning models by increasing the task alignments. Their key assumption is that standard MTL can result in catastrophic forgetting of pretrained knowledge and lead to local task optimum in the finetuning stages. To avoid this, the paper proposed a method to jointly optimize the task losses and gradient alignments between tasks. Their key contributions are:
- The paper empirically examined the question of whether worse alignments of task gradients can lead to negative transfer, though extensive experiments on multilingual tasks.
- Based on such observation, the paper proposes a simple and effective optimization method for MTL that jointly optimizes the task losses and gradient alignments between tasks.

**Summary Of The Review:**

Overall, I think the paper's merits outweigh its drawbacks. It provides some important questions and findings to multi-task learning. Therefore I would recommend accept the paper.

---

> ### Author Response · Authors · 2021-11-15
> **Response to Reviewer ZkcB (2/2)**
>
>
> **[Q3]** In the synthetic experiment, instead of only showing the loss landscape of a single initialization point, it would be more convincing if such patterns hold for random initialization.
> - In the revision, we additionally visualize the learning trajectories starting from different initializations (Figure 9,10,11 in Appendix D). The same patterns hold for diverse initializations, as expected.
> ---
> **[Q4]** Average performance of each language doesn't consider the differences of the number of train/eval examples for different languages. Typically, we observed most MTL methods hurt for high-resource languages while improving on low-resource languages. Simply averaging across different languages leads to a discrepancy between MTL model training and results reporting, as the loss is optimized as the sum of the example losses in training time. Therefore, simply averaging the performance across languages can be a misleading performance report as the actual empirical effectiveness per example may differ. But I guess this issue exists in most MTL papers.
> - We already take this discrepancy into account by using the sampling distribution $p_t \propto (N_t)^{1/5}$ at training time. Because this sampling distribution is flatter than $p_t \propto N_t$, it can roughly balance the imbalance between high and low resource languages. For this reason, this sampling distribution is commonly used in MTL for NLP [1,2,3]. Reporting average performance is reasonable under this sampling distribution.
> ---
> **[Q5]** In Table 5 & 6, it would be good to include the single-task learning (STL) performance on zero-shot cross-lingual experiments.
> - In Table 5 and 6, we have included the results of STL on zero-shot cross-lingual experiments. Again, **our Sequential Reptile largely outperforms STL on most of the languages**.
> ---
> **[Q6]** More experiments on some hyper-parameters could lead to better understanding of the algorithms, especially on inner and outer learning rates and inner gradient steps.
> - In Figure 7(b), we already plot EM score and average cosine similarity between task gradients as a function of inner learning rate. It seems important to carefully select the inner-learning rate to find a good tradeoff between minimizing MTL loss and maximizing cosine similarity. The best performing value is $3\cdot 10^{-5}$, which is the default LR for BERT finetuning.
> - Moreover, we **additionally include** Figure 8(b) in the revision (Appendix D) that **plots EM score vs. the number of inner steps**. Sequential Reptile performs similarly with the varying number of inner-gradient steps.
>
> ---
>
>
> # References
> [1] Arivazhagan, Naveen, et al. "Massively multilingual neural machine translation in the wild: Findings and challenges." arXiv preprint arXiv:1907.05019 (2019).
>
> [2] Wang, Zirui, Zachary C. Lipton, and Yulia Tsvetkov. "On Negative Interference in Multilingual Language Models." Proceedings of the 2020 Conference on Empirical Methods in Natural Language Processing (EMNLP). 2020.
>
> [3] Wang, Zirui, et al. "Gradient Vaccine: Investigating and Improving Multi-task Optimization in Massively Multilingual Models." International Conference on Learning Representations. 2021.

---

> > ### Comment · Reviewer_ZkcB · 2021-12-01
> > **additional comment**
> >
> > I have went through all the comments and thanks authors for the detailed responses. Although I like the simplicity of the proposed approach and their interpretation & empirical verification on the alignment of task gradients, I decided to not raise my score based on the following experimental results:
> >
> > 1. In the GLUE benchmark, it seems the proposed algorithm only has marginal improvements against the standard MTL setting. It has meaningful improvements than the baseline in 4 benchmarks, while is on-par or worse on the other 4 benchmarks. Also it would be good to check out why reptile doesn't work in both GLUE and image classification tasks.
> >
> > 2. In the synthetic experiments, it seems reptile leads much better solution than sequential reptile. Any explanation on that?

---

> > > ### Author Response · Authors · 2021-12-02
> > > **Thank you for the response**
> > >
> > > Thank you for the response. We would like to kindly answer your comments and questions.
> > >
> > > **[Q1]**. In the GLUE benchmark, it seems the proposed algorithm only has marginal improvements against the standard MTL setting. It has meaningful improvements than the baseline in 4 benchmarks, while is on-par or worse on the other 4 benchmarks.
> > > - We want to emphasize that in GLUE benchmark, Sequential Reptile shows better performance than the base MTL in terms of **average accuracy**. In achieving this superior performance, it only requires **half of the computational costs** than the base MTL in terms of the number of forward and backward computations.
> > > - Also, it is only this specific experiment that shows somewhat marginal improvements. In all the other experiments including multilingual MTL, zero-shot cross-lingual transfer, and even the large-scale image dataset experiments, our Sequential Reptile shows huge improvements than all the various baselines. Again, in achieving this superior performance over the various settings, the computational cost is much lower than the baselines.
> > > - Therefore, in terms of performance and computational efficiency, we **strongly** believe that our **Sequential Reptile is competitive enough and should be highlighted at the conference**. All the experimental results strongly support the effectiveness and efficiency of our method.
> > >
> > > **[Q2]**. Also it would be good to check out why Reptile doesn't work in both GLUE and image classification tasks.
> > > - Recall that Reptile performs task-specific adaptations and then aggregates the evolved weights by performing a meta-update. This procedure is not problematic for multilingual tasks where we can share the whole model across the tasks. However, this aggregation step causes a problem when the label space is not shared across the tasks, i.e. when we need to assume different task-specific heads on top of the shared feature extractor as with GLUE and image classification tasks.
> > > - The problem is that if we need to assume different task-specific heads and Reptile performs the aggregation only with the feature extractor weights, then Reptile gets to have a discrepancy between the shared feature extractor and the task-specific heads right after a meta-update. The evolved feature extractors are averaged out to be shared across tasks, but the task-specific heads remain task-specific. This discrepancy is the main source of performance degradation for GLUE and image classification tasks.
> > > - This is not an issue for meta-learning settings because we can further task-specifically fine-tune the feature extractor at the meta-test time. For multi-task learning settings, however, we want to have a single model for prediction without any task-specific adaptations, so we cannot avoid such discrepancy.
> > > - In contrast, Sequential Reptile less suffers from the discrepancy issue since it keeps a single inner-trajectory shared across the tasks. Since the backbone is always shared across the tasks, the meta-update (i.e. aggregation step) does not significantly change the backbone weights during the training. As a result, it is easy for the backbone network and classifier head to align during the training.
> > >
> > > **[Q3]**. In the synthetic experiments, it seems Reptile leads to a much better solution than Sequential Reptile. Any explanation on that?
> > > - Firstly, in this synthetic experiment, it is **not straightforward to define a test loss** and thus it is hard to tell which convergence point is “better” than the other in terms of generalization. Although Reptile converges to a point in the middle of three local optima, we cannot tell whether the point would generalize better than the point obtained by Sequential Reptile.
> > >
> > > - Rather, what we want to emphasize from this synthetic experiment is Figure 3(h). In this figure, the cosine similarity, the degree of inter-task gradient alignment, is largely different between Reptile ($\approx -0.5$) and Sequential Reptile ($\approx 0. 3$). The results demonstrate that Reptile **does not consider** the tradeoff between minimizing MTL loss and inter-task gradient alignments (see Figure 1). This is problematic as we consistently observe from our experiments that the gradient misalignment can lead to severe negative transfer and catastrophic forgetting.

---

> ### Author Response · Authors · 2021-11-15
> **Response to Reviewer ZkcB (1/2)**
>
> We sincerely appreciate your constructive comments. We respond to the individual comments below:
>
>
> **[Q1]** Is the negative transfer actually caused by the misalignments of task gradients?
> - **The relationship is causal** because comparing the performance of Reptile and Sequential Reptile over various settings is actually a **randomized controlled experiment**. We can mathematically show (see appendix C) that the only difference between Reptile and Sequential Reptile is its ability of inter-task gradient alignment. Therefore, by choosing between Reptile and Sequential Reptile, we can carefully control the effect of all other explanatory factors besides the inter-task gradient alignment. We will make it clear in the revision.
>
> **[Q2]** The paper's findings would be much more generalizable if the experiments are also performed in the MTL setting of a single language, i.e., in a standard MTL setting.
> - Thank you for your insightful comment. Following your suggestion, **we have conducted MTL experiments on the GLUE dataset as well as multi-task image classifications**. They are different from the multilingual learning task in that we need a separate task-specific head (classifier or regressor) for each task.
> - Again, **Sequential Reptile largely outperforms the relevant baselines**, demonstrating its effectiveness for solving general MTL problems. We have included the new experimental results in the revision (Appendix E).
> ---
> **< General MTL setting 1: GLUE >**
>
> | **Method**       |    **CoLA**   |    **MNLI**   |    **MRPC**   |    **QNLI**   |    **QQP**    |    **RTE**    |    **SST2**   |    **STSB**   |    **Avg.**    |
> |--------------|:---------:|:---------:|:---------:|:---------:|:---------:|:---------:|:---------:|:---------:|:---------:|
> | MTL          |   80.82   | **83.47** |    82.1   | **90.50** | **90.16** |   68.23   |   90.13   | **89.73** |   84.39   |
> | Reptile      |   82.07   |   78.81   |    81.3   |   89.29   |   87.57   |   72.56   |   88.76   |   88.55   |   83.61   |
> | **Seq. Reptile** | **82.35** |   83.02   | **83.30** | **90.50** |    89.4   | **73.28** | **92.31** |    89.4   | **85.44** |
>
> ---
> **< General MTL setting 2: Image Classification >**
>
> | **Method**       | **TIN-1**     | TIN-2     | CIFAR100  | Dogs      | Aircraft  | CUB       | FMNIST    | SVHN      | Avg.      |
> |--------------|:---------:|:---------:|:---------:|:---------:|:---------:|:---------:|:---------:|:---------:|:---------:|
> | MTL          |     56.26 |      53.40 |     54.43 |     33.44 |     44.97 |     29.48 | **90.19** |     87.70 |     56.22 |
> | Reptile      |     23.56 |     23.28 |      9.64 |      12.2 |     10.02 |      6.61 |     60.87 |     20.86 |     20.88 |
> | **Seq. Reptile** | **58.12** | **56.83** | **57.46** | **38.05** | **59.55** | **35.96** |     87.44 | **88.86** | **60.28** |

---

> ### Author Response · Authors · 2021-11-25
> **A Gentle Reminder**
>
> Dear reviewer
>
> We sincerely appreciate your efforts in reviewing our paper, and your constructive comments. We have responded to your comments, faithfully reflected them in the revision, and provided additional experimental results that you have requested. Could you please go over our responses and the revision since end of the final discussion phase is approaching? Please let us know there is anything else we need to clarify or provide.
>
> Thanks, authors.

---

### Author Response · Authors · 2021-11-15
**Summary of the Revision**

We really appreciate all the reviewers for their constructive comments. We have responded to the individual comments from the reviewers below, and believe that we have successfully responded to most of them. We have included the discussions and results of the suggested experiments in the revision. Here we briefly summarize the updates we have made to the revision:

- We have included **additional experiments on GLUE dataset and image classification** in Appendix E, as suggested by Reviewer ZkcB and vCuD.

- We have included the **additional baseline** which learns **language invariant representation** for all the experiments, as suggested by Reviewer vCuD.

- We have included **wall-clock time comparison** between Sequential Reptile and MTL in Appendix D, as suggested by Reviewer gNwa.

- We have included **additional visualization (Figure 9,10,11) of learning trajectory with different initialization**, as suggested by Reviewer ZkcB.

- We have included Figure 8(b) in Appendix which shows the **EM performance with varying the number of inner steps**.

---

### Author Response · Authors · 2021-11-22
**The end of the discussion phase approaching**

Dear Reviewers,

Could you please go over our responses and the revision since we can have interactions with you only by this Monday (22nd)? We have responded to your comments and faithfully reflected them in the revision, and provided additional experimental results that you have requested. We sincerely thank you for your time and efforts in reviewing our paper, and your insightful and constructive comments.

Thanks, Authors

---

### Decision · Program_Chairs · 2022-01-20

**Decision:**

Accept (Poster)

**Comment:**

This paper presents a gradient alignment approach to alleviate negative transfer and catastrophic forgetting for multitask and cross lingual learning. Experiments on many domains and datasets demonstrate the efficacy of the proposed approach.

All reviewers agree that the simplicity of the proposed method is a strength of the paper and the experiments are promising. They have suggestions to improve the experiments section, which I believe the authors have addressed in their rebuttal by adding GLUE, image classification, and statistical significance tests, among others.

I recommend accepting this paper for ICLR.